# A two-step activation mechanism enables mast cells to differentiate their response between extracellular and invasive enterobacterial infection

Christopher von Beek [1], Anna Fahlgren[2], Petra Geiser [1], Maria Letizia Di Martino [1], Otto Lindahl [1], Grisna I. Prensa [1], Erika Mendez-Enriquez [1], Jens Eriksson [1], Jenny Hallgren [1], Maria Fällman [2], Gunnar Pejler[1] ✉ & Mikael E. Sellin [1,3] ✉

Mast cells localize to mucosal tissues and contribute to innate immune defense against infection. How mast cells sense, differentiate between, and respond to bacterial pathogens remains a topic of ongoing debate. Using the prototype enteropathogen *Salmonella* Typhimurium (*S*.Tm) and other related enterobacteria, here we show that mast cells can regulate their cytokine secretion response to distinguish between extracellular and invasive bacterial infection. Tissue-invasive *S*.Tm and mast cells colocalize in the mouse gut during acute *Salmonella* infection. Toll-like Receptor 4 (TLR4) sensing of extracellular *S*.Tm, or pure lipopolysaccharide, causes a modest induction of cytokine transcripts and proteins, including IL-6, IL-13, and TNF. By contrast, type-III-secretion-system-1 (TTSS-1)-dependent *S*.Tm invasion of both mouse and human mast cells triggers rapid and potent inflammatory gene expression and >100-fold elevated cytokine secretion. The *S*.Tm TTSS-1 effectors SopB, SopE, and SopE2 here elicit a second activation signal, including Akt phosphorylation downstream of effector translocation, which combines with TLR activation to drive the full-blown mast cell response. Supernatants from *S*.Tm-infected mast cells boost macrophage survival and maturation from bone-marrow progenitors. Taken together, this study shows that mast cells can differentiate between extracellular and host-cell invasive enterobacteria via a two-step activation mechanism and tune their inflammatory output accordingly.

Mast cells (MCs) are innate immune cells found all over the body, but particularly enriched in barrier tissues, including the skin, the lung and intestinal mucosae. In addition to their well-known involvement in allergy, MCs take part in the host response to a wide variety of infections[1–5]. Their strategic localization makes them frequent early responders to pathogens that disrupt epithelial linings.

In analogy with other immune cells, MCs can sense bacterial pathogen-associated molecular patterns (PAMPs) via Toll-like receptors (TLRs), e.g., TLR2 and TLR4, which detect cell wall components

[1]Department of Medical Biochemistry and Microbiology, Uppsala University, Uppsala, Sweden. [2]Department of Molecular Biology, Laboratory for Molecular Infection Medicine Sweden (MIMS), Umeå Centre for Microbial Research (UCMR), Umeå University, Umeå, Sweden. [3]Science for Life Laboratory, Uppsala, Sweden. ✉e-mail: gunnar.pejler@imbim.uu.se; mikael.sellin@imbim.uu.se

and lipopolysaccharide (LPS), respectively[6–9]. Other receptors present in certain MC subsets include the peptidoglycan sensor Nod1[10], and the Mas-related G-protein-coupled receptor X2 (MRGPRX2) that detects bacterial quorum sensing molecules[11]. Moreover, studies of diverse infections have demonstrated that MCs can detect assault by bacterial cytolytic toxins[12–16]. Based on the latter studies, a common theme emerges, in which MCs respond vigorously to sublytic membrane perturbation that precedes toxin-mediated lysis[16,17]. It is unclear whether this mode of sensing also extends to other membrane-interfering events, such as the docking of bacterial secretion machineries to the plasma membrane.

Upon IgE-crosslinking of FcεRI receptors, or in response to some infectious stimuli, MCs degranulate rapidly and release pre-stored mediators, including histamine, proteases, and pro-inflammatory cytokines[18]. Activated MCs also exhibit de novo biosynthesis of lipid mediators secreted within minutes, as well as cytokines and chemokines reaching detectable levels within hours[19]. The processes of MC degranulation and cytokine/chemokine production and secretion may occur in parallel but often appear uncoupled during infection. It remains poorly understood how MCs coordinate their different modes of bacterial sensing, and tune the nature and magnitude of their response to match the stimulus.

An additional controversy concerns the capacity MCs have to internalize bacteria. It has been proposed that bacteria, in contrast to viruses, are not internalized by MCs, and that this may explain why specifically viral infection generates a potent type I interferon response in MCs[6]. However, other studies offer contradicting results, showing that *Staphylococcus aureus* can be internalized by murine bone marrow-derived MCs (BMMCs), human cultured MC models and nasal polyp MCs in vivo[20–23]. Further, evidence exists for some degree of MC uptake/phagocytosis also of other bacteria, e.g., non-opsonized or serum-opsonized *Escherichia coli* (*E. coli*)[24,25], *Streptococcus faecium*[26], and *Chlamydia trachomatis*[27]. Thus far, no consensus has emerged on how the bacterial location affects the subsequent MC response.

Enterobacterial infections of the intestine represent one of the most prevalent classes of infectious diseases, with estimates of >600 million yearly disease cases[28]. These infections are caused by closely related gram-negative bacteria within e.g., the *Escherichia*, *Shigella*, *Yersinia*, and *Salmonella* genera. *Salmonella enterica* serovar Typhimurium (*S.*Tm) is a globally significant pathogen and a model bacterium for studies of enterobacterial pathogenesis[29]. *S.*Tm employs flagella to swim towards the intestinal epithelium and a type-three-secretion system (TTSS-1) to translocate effectors into targeted host cells. These effectors activate multiple Rho- and Arf-GTPases and formins, and induce actin-dependent bacterial uptake[29,30]. By this means, *S.*Tm efficiently invades intestinal epithelial cells, but also many other non-phagocytic and phagocytic cell types[31–35]. Both *S.*Tm and the other related enterobacteria can, however, also prevail in the extracellular environment, raising the question how our immune cells react to such disparate microbial behaviors.

Here, we have explored the MC interaction with *S.*Tm and related enterobacteria across a panel of experimental models, combining bacterial genetics with readouts for MC activation. We find that TTSS-1-proficient *S.*Tm efficiently invade MCs, whereas *S.*Tm grown under non-TTSS-1-inducing conditions, or genetically deleted for TTSS-1 components, do not. Remarkably, the MCs can tune their cytokine response to accomplish slow and low-level cytokine production when detecting extracellular enterobacteria, but swift and full-blown cytokine production in response to invasive *S.*Tm strains. This can be explained by a two-step MC activation mechanism, whereby extracellular bacteria only fuel a TLR signal, while for invasive *S.*Tm this signal combines with additional signal(s), elicited by the TTSS-1 effectors SopB/SopE/SopE2 and involving Akt pathway stimulation. This illustrates how MCs can cater their cytokine secretion response to inform their surroundings about the virulence behavior of a bacterial intruder.

## Results

### Mast cells and *Salmonella* coexist in the infected murine gut

To investigate the distribution of MCs in the bacterium-infected gut, we utilized a mouse model of *Salmonella* enterocolitis[36]. Mice were infected per oral gavage with $3.0–7.5 \times 10^6$ colony-forming units (CFUs) of *S.*Tm[wt] SL1344 for 48 h. Toluidine blue staining of cecal tissue sections confirmed the presence of MCs in the mucosa and submucosa of uninfected controls and *S.*Tm-infected animals (Fig. 1a, b). When co-staining for *S.*Tm LPS and avidin (for MCs), we reproduced the MC tissue distribution in uninfected controls (Fig. 1c), and detected MCs close to *S.*Tm-infested regions of the intestinal epithelium (Fig. 1d, e and Supplementary Fig. 1A), as well as in the deeper mucosa (Fig. 1e and Supplementary Fig. 1A), of infected animals. The total number of tissue MCs showed a modest elevation upon infection (Fig. 1f), predominantly explained by increased mucosal MC numbers (Fig. 1g, h). Next, we quantified transcript levels for MC-specific proteases in cecal tissue (Fig. 1i). mRNAs encoding the mucosal MC proteases Mcpt1 and Mcpt2, as well as the connective tissue MC protease Mcpt4, were relatively highly expressed, while median transcript levels for other connective tissue MC proteases (Mcpt5, Mcpt6, Cpa3) approached the detection limit (Fig. 1i). When comparing control and *S.*Tm-infected sample groups, a trend towards elevated *Mcpt2* levels was noted in the latter. Hence, MCs lodge within the mucosa and submucosa of intestinal tissue prior to *S.*Tm infection, and enrich further in the mucosa by 48 h post infection (p.i.). Notably, in infected mice exhibiting significant inflammation, we also detected avidin+ staining in the epithelium-proximal lumen, which at this stage harbors high densities of *S.*Tm mixed with extruded epithelial cells and transmigrated myeloid and lymphoid cells (Fig. 1j and Supplementary Fig. 1A[37,38]). Frequently, luminal avidin+ material did not colocalize with a distinct DAPI+ nuclear morphology. This shows that MCs come in close contact with invasive *S.*Tm in the superficial gut mucosa (distances quantified in Supplementary Fig. 1B), and can also enter into and succumb in the *S.*Tm-filled lumen. Further analyses of cecal tissue from a later chronic infection stage (42 days p.i.) again revealed the presence of submucosal, mucosal, and luminal MCs (Supplementary Fig. 1C–J), with MC numbers broadly similar to at 48 h p.i. (Supplementary Fig. 1J). However, at this stage the tissue presented with a more variable appearance, local patches of *S.*Tm colonization, and fewer obvious examples of MC–*S.*Tm proximity (Supplementary Fig. 1G–I). We conclude that tissue-invasive *S.*Tm and MCs encounter each other in the infected gut mucosa and lumen, particularly during acute infection.

### TTSS-1-proficient invasive *Salmonella* trigger cytokine secretion from mast cells in the absence of degranulation

To assess how MCs respond to tissue-invasive *S.*Tm, we exposed bone marrow-derived MCs (BMMCs) to *S.*Tm[wt] at varying multiplicities of infection (MOI) and time frames. The *S.*Tm inoculum was cultured to promote TTSS-1 expression and invasiveness (see "Methods" and Supplementary Fig. 2A, B[32,39]), similar to in the gut. BMMCs responded to the infection by secretion of IL-6 protein in a MOI- and time-dependent manner (Supplementary Fig. 2C, D). This response was most vigorous at MOI ~25–50, and diminished again at higher MOIs (Supplementary Fig. 2C), which may be explained by dose-dependent toxicity at excessive bacterial loads. IL-6 secretion was noted from 2 h p.i. and increased considerably by 3–4 h p.i. (Supplementary Fig. 2D), whereas we detected significantly elevated *Il6* transcript levels already by 1 h p.i. (Supplementary Fig. 2E). However, only minimal BMMC degranulation could be detected within 1–24 h p.i., as assayed by β-hexosaminidase release (Supplementary Fig. 2F–H). We also reassessed MC degranulation using another TTSS-1-proficient *S.*Tm strain background (ATCC 14028). Again, neither *S.*Tm[14028 wt], nor a *S.*Tm[14028 ΔsptP] mutant that lacks the SptP effector previously suggested to block MC degranulation[40], elicited above-background β-hexosaminidase release during the first hour (Supplementary Fig. 2I), but

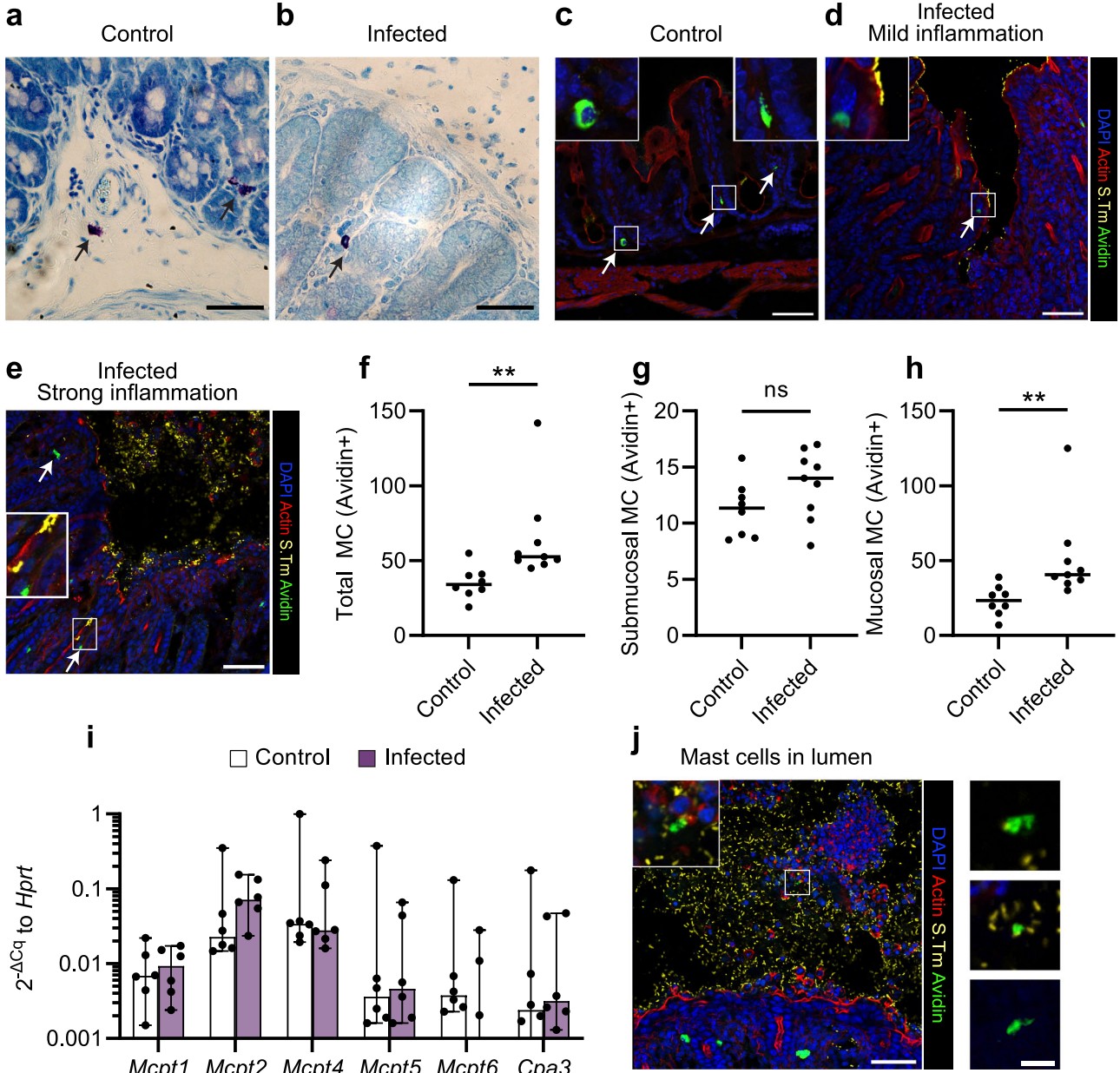

**Fig. 1 | Mast cells are found in the *Salmonella*-infected cecum and come in close contact with invading bacteria. a**, **b** Toluidine blue-stained tissue sections of cecum from uninfected mice and 48 h after *S*.Tm$^{wt}$ SL1344 infection. Arrows indicate different MC locations such as submucosa (**a**, bottom) and mucosa (**a**, top, **b**). Scale bars: 50 μm. **c**–**e** Representative IF images for cecum of uninfected mice, or infected mice in different stages of inflammation as indicated in the panel headings. Arrows indicate the position of MCs in healthy mucosa and submucosa (**c**), close to the epithelial layer (**d**) or close to invading bacteria (**e**). Scale bars: 50 μm. **f**–**h** Quantification of avidin+ cells per cecum section as total numbers (**f**), submucosal MCs (**g**) or mucosal MCs (**h**). Every dot indicates the mean of at least three sections for one mouse, *n* = 8 (Control), 9 (Infected). Horizontal lines display median, and the significance of two-sided Mann−Whitney *U* test is shown.

**P < 0.01; ns nonsignificant. Exact *P* values given in source data. **i** RT-qPCR analysis of total cecum tissue for mast cell protease transcripts relative to *Hprt* (2$^{-\Delta Cq}$). A threshold of expression derived from Cq values < 38 was chosen. Note that some values fall under the threshold and are therefore not visible. One dot represents transcript levels in one mouse, *n* = 6. Bars indicate median ± 95% confidence intervals. No significance (*P* > 0.05) was detected for any comparison by two-sided Mann−Whitney *U* test. **j** Avidin+ cells present in the lumen of infected cecum tissue sections. Representative overview image (scale bars: 50 μm) and magnified images from avidin and anti-*S*.Tm-co-staining. Images are representative from sections of all mice; the number of MCs in the lumen are however higher in mice with strong inflammation, such as in the selected images. Scale bars: 10 μm.

the calcium ionophore A23187 (positive control) did (Supplementary Fig. 2F, I). Measurement of the percentage CD63$^{high}$ BMMCs at 1–24 h p.i., as an alternative readout for degranulation[41,42], corroborated these results (Supplementary Fig. 2J–I). Hence, exposure to TTSS-1-primed *S*.Tm$^{wt}$ elicits swift *Il6* transcription and IL-6 protein secretion from BMMCs in the absence of degranulation.

Bacterial recognition by MCs has often been accredited to TLRs[6,9]. However, work by us and others have in addition shown that sublytic

levels of bacterial pore-forming toxins can elicit secretion of cytokines, including IL-6, from MCs[14–16]. We therefore asked if MC activation by *S*.Tm could be ascribed to (i) classical recognition of bacterial PAMPs, (ii) membrane-perturbing effects of the TTSS-1 translocon (in analogy to pore-forming toxins), or (iii) alternative mechanism(s). To address this question, we infected parallel BMMC cultures (MOI 50, 4 h) with either S.Tm$^{wt}$, a mutant lacking the structural TTSS-1 component InvG (S.Tm$^{\Delta invG}$), or one that retains the TTSS-1 apparatus and the

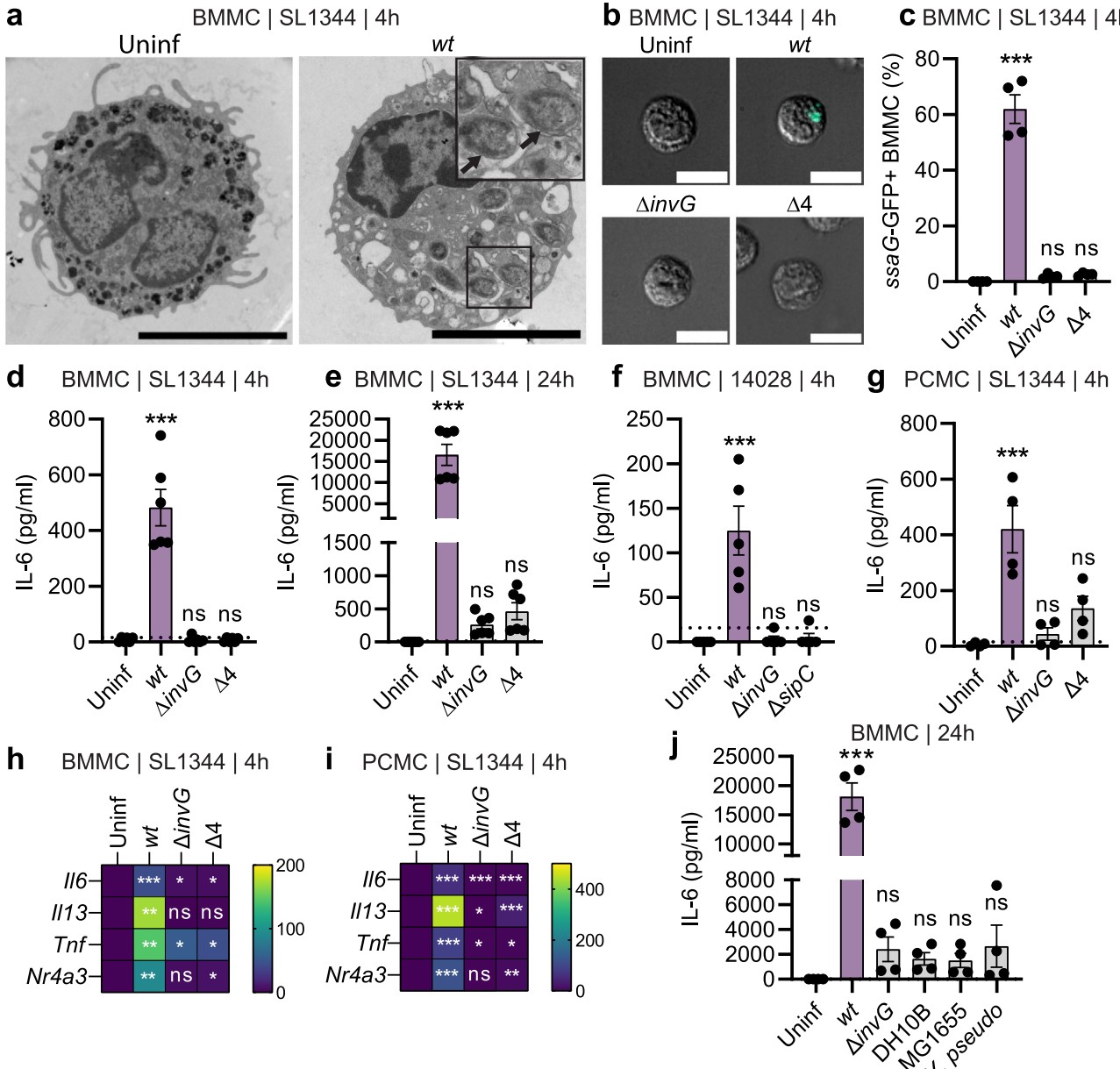

**Fig. 2 | Mast cells mount a potent immunomodulatory response to *Salmonella*, which is triggered by TTSS-1 effectors. a** Representative TEM images of BMMCs infected with *S*.Tm^wt^ SL1344 for 4 h, as well as uninfected BMMCs. Arrows indicate intracellular bacteria. Scale bar: 5 μm. **b, c** Representative 25 ×25 μm images (**b**) and quantification by flow cytometry (**c**) of BMMCs, infected with MOI 50 of S.Tm^wt^ SL1344 or the indicated TTSS-mutants for 4 h. GFP signal and quantification show vacuolar *S*.Tm within BMMCs. *n* = 4. **d–g** Similar conditions as above but analysis of secreted IL-6 after 4 h (*n* = 6) (**d**) and 24 h (*n* = 6) (**e**). **f** Similar setup as in (**d**), but *S*.Tm14028 strains were used (*n* = 5). **g** Similar setup as in (**d**), but PCMCs were used

(*n* = 4). **h, i** Heatmap for RT-qPCR-quantified transcript levels for *Il6*, *Il13*, *Tnf* and *Nr4a3* in BMMCs (**h**) and PCMCs (**i**), infected for 4 h by the indicated *S*.Tm SL1344 strains. **j** Secreted IL-6 levels from BMMCs infected with MOI 50 of S.Tm^wt^ and *S*.Tm^ΔinvG^ SL1344 as well as *E. coli* DH10B, *E. coli* MG1655 and *Y. pseudotuberculosis* for 24 h (*n* = 4). Every experiment was performed 2–3 times and mean ± SEM of pooled biological replicates is shown. Uninfected cells were used for statistical comparisons by one-way ANOVA and Dunnett's posthoc test to all other groups. ***$P < 0.001$; ns−nonsignificant. Exact p values given in source data. For *S*.Tm ATCC 14028 infections, a Δ*malX* strain was used as *wt*.

membrane-interacting translocon, but lacks the host cell invasion effectors SipA, SopB, SopE, and SopE2 (*S*.Tm^Δ4^). First, transmission electron microscopy (TEM) revealed that approximately half of all *S*.Tm^wt^-infected BMMCs harbored intracellular bacteria (exemplified in Fig. 2a). The introduction of an *ssaG*-GFP reporter[43] (validated against a constitutive reporter; Supplementary Fig. 2B) allowed us to assess if *S*.Tm^wt^ and the mutant strains could invade and establish an intracellular niche within MCs. In agreement with the TEM, *S*.Tm^wt^ efficiently invaded MCs, whereas the *S*.Tm^ΔinvG^ and *S*.Tm^Δ4^ strains were non-invasive (Fig. 2b, c). Strikingly, while *S*.Tm^wt^ elicited prompt secretion of IL-6 (-500 pg/ml), IL-13 (-50 pg/ml), and TNF (-50 pg/ml) from the

BMMCs, *S*.Tm^ΔinvG^ and *S*.Tm^Δ4^ failed to do so (Fig. 2d and Supplementary Fig. 2M–N). Following a longer 24-h infection, *S*.Tm^ΔinvG^ and *S*.Tm^Δ4^ did elicit above-background levels of IL-6 secretion, but still vastly lower than the >15,000 pg/ml noted for *S*.Tm^wt^ (Fig. 2e). These findings were generalizable also to infection of BMMCs with invasive and noninvasive *S*.Tm^14028^ strains (Fig. 2f), and to infections of murine peritoneal cell-derived mast cells (PCMCs) (Fig. 2g). Moreover, *S*.Tm^wt^ induced markedly higher levels of *Il6*, *Il13*, *Tnf*, and *Nr4a3* (a MC transcription factor responsive to other stimuli[16,44];) transcripts than *S*.Tm^ΔinvG^ and *S*.Tm^Δ4^, both in BMMCs (Fig. 2h; individual data points shown in Supplementary Fig. 3A–D) and in PCMCs (Fig. 2i and

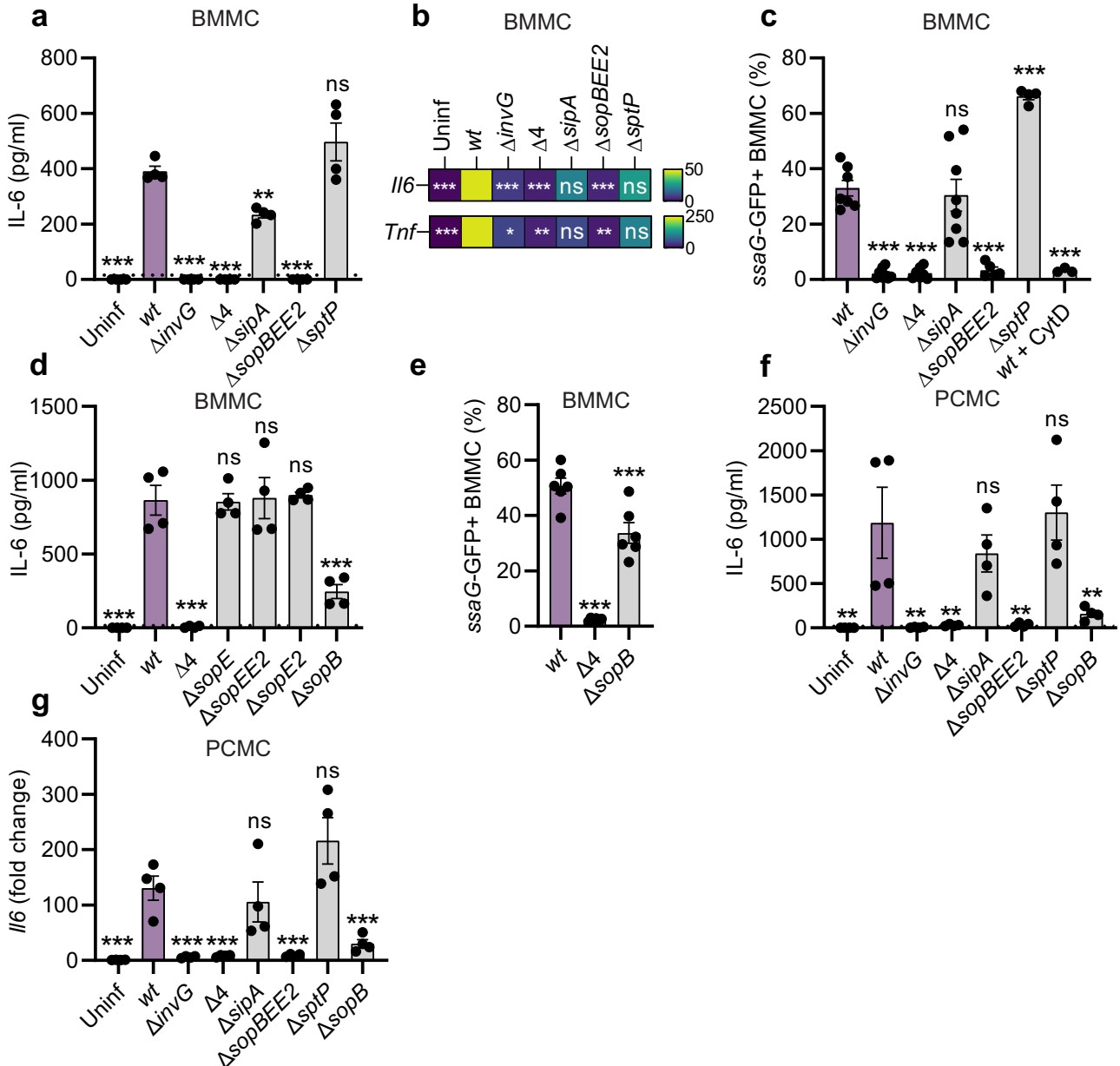

**Fig. 3 | The TTSS-1 effectors SopB, SopE, and SopE2 induce mast cell cytokine expression and secretion upon *Salmonella* infection. a–g** BMMCs (**a–e**) or PCMCs (**f, g**), infected with MOI 50 of *S*.Tm$^{wt}$ SL1344 or the indicated TTSS-mutants for 4 h. **a** (*n* = 4), **d** (*n* = 4), and **f** (*n* = 4) show IL-6 secretion, **b** shows a heatmap for RT-qPCR-quantified transcript levels of *Il6* and *Tnf* in BMMCs, **g** shows *Il6* transcript levels in PCMCs (*n* = 4). **c** (*n* = 7, 8, 7, 8, 5, 4, 3, respectively, for groups from left to right) and **e** (*n* = 6) show percentage of BMMCs harboring vacuolar (*ssaG*-GFP + ) *S*.Tm. Every experiment was performed 2–4 times and mean ± SEM of pooled biological replicates is shown. *S*.Tm$^{wt}$-infected cells were used for statistical comparisons by one-way ANOVA and Dunnett's posthoc test to all other groups. *$P < 0.05$; **$P < 0.01$; ***$P < 0.001$; ns nonsignificant. Exact *P* values given in source data.

Supplementary Fig. 3G–J). It should be noted that some other transcripts, including *Il1b* and *Nlrp3*, were upregulated by all strains (Supplementary Fig. 3E, F, K, L), suggesting the existence of several transcriptional programs that may react to different cues.

Finally, the BMMC response to *S*.Tm was contrasted to three other noninvasive enterobacteria, namely the *E. coli* strains DH10B and MG1655, and wild-type *Yersinia pseudotuberculosis* (*Y. pseudotuberculosis*$^{wt}$; has a TTSS apparatus, but uses it to prevent host cell uptake[45]). Strikingly, all three *E. coli/Y. pseudotuberculosis* strains elicited modest levels of IL-6 secretion, and virtually undetectable IL-13 secretion, thereby phenocopying the *S*.Tm$^{ΔinvG}$ strain (Fig. 2j and Supplementary Fig. 2O). By sharp contrast, *S*.Tm$^{wt}$ again elicited a vigorous cytokine response (Fig. 2j and Supplementary Fig. 2O). We conclude that TTSS-1-proficient invasive *S*.Tm (*S*.Tm$^{wt}$) trigger a potent

transcriptional and cytokine secretion response in MCs, that is neither recapitulated by related noninvasive enterobacteria, nor by *S*.Tm strains genetically attenuated for invasive behavior (*S*.Tm$^{ΔinvG}$, *S*.Tm$^{Δ4}$), even when these retain the membrane-interacting TTSS translocon (*S*.Tm$^{Δ4}$, *Y. pseudotuberculosis*$^{wt}$).

## The TTSS-1 effectors SopB, SopE, and SopE2 promote mast cell cytokine secretion upon *Salmonella* infection

Next, we surveyed the impact of individual *S*.Tm TTSS-1 effectors. Deletion of SipA (*S*.Tm$^{ΔsipA}$) or SptP (*S*.Tm$^{ΔsptP}$) marginally decreased, respectively marginally increased (nonsignificant), the levels of *S*.Tm-induced IL-6 secretion from BMMCs by 4 h p.i. (Fig. 3a). By contrast, simultaneous removal of SopB, SopE, and SopE2 (*S*.Tm$^{ΔsopBEE2}$) phenocopied the *S*.Tm$^{Δ4}$ and *S*.Tm$^{ΔinvG}$ strains (i.e., minimal IL-6 secretion at

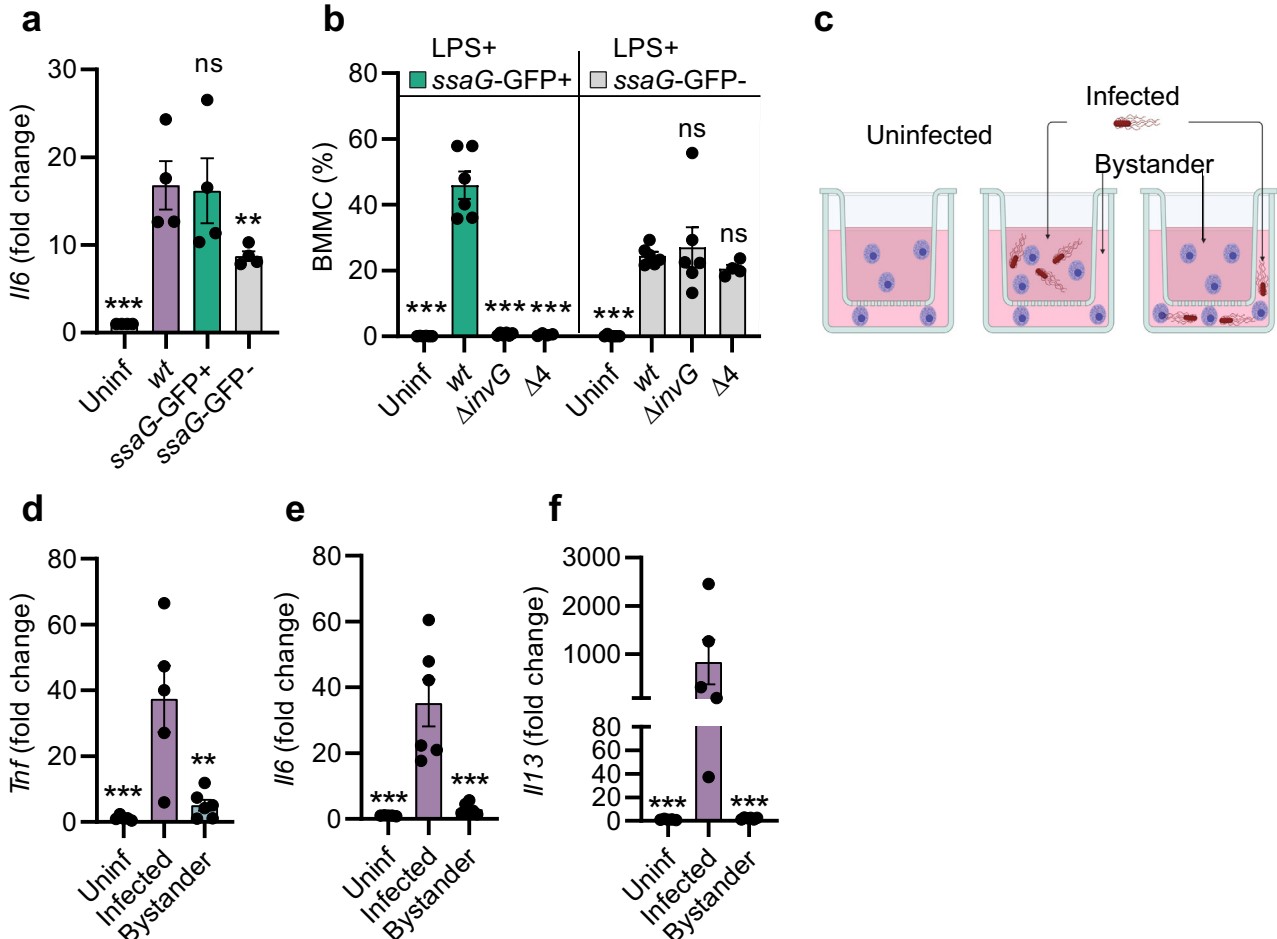

**Fig. 4 | *Salmonella*-invaded mast cells are the main source of cytokines.**
**a** BMMCs were infected with *S*.Tm$^{wt}$ SL1344 carrying the p*ssaG*-GFP reporter for 4 h and sorted to enrich the *ssaG*-GFP- population. *Il6* transcript levels for both fractions as well as uninfected and unsorted *S*.Tm-infected BMMCs are shown (*n* = 4).
**b** BMMCs were infected with *S*.Tm$^{wt}$ SL1344 for 4 h and stained for *Salmonella* LPS. Relative population sizes of MCs positive for LPS and/or vacuolar (*ssaG*-GFP + ) *S*.Tm (*n* = 4 for "Δ4", *n* = 6 for others). **c** Experimental setup for analysis of the source of soluble factors secreted by MCs. **d**–**f** BMMCs were infected with *S*.Tm$^{wt}$ in one compartment while separated from BMMCs in the other compartment not coming in direct contact with the bacteria. RT-qPCR-quantified transcript levels for *Tnf*, *Il6* and *Il13*, respectively, in the aforementioned transwell compartments (*n* = 6 for "Bystander", *n* = 5 for others). Experiments were performed 2–3 times and mean ± SEM of pooled biological replicates is shown. **a**, **c**–**f** *S*.Tm$^{wt}$-infected cells were used for statistical comparison by one-way ANOVA and Dunnett's post hoc test to all other groups. **b** Two-way ANOVA with Dunnett's posthoc test was used to compare *S*.Tm$^{wt}$-infected cells within each subpopulation to all other groups. **P* < 0.01; ****P* < 0.001; ns nonsignificant. Exact p values given in source data. Panel **c** was created with BioRender.com.

4 h p.i.) (Fig. 3a). Analysis of *Il6* and *Tnf* transcripts further corroborated these findings (Fig. 3b and Supplementary Fig. 4A, B). To assess how the effectors influenced BMMC invasion, we again exploited *ssaG*-GFP reporter strains (Fig. 3c). Flow cytometry showed that *S*.Tm$^{wt}$ and *S*.Tm$^{ΔsipA}$ invaded BMMCs to an equal degree, while *S*.Tm$^{ΔsopBEE2}$ failed to establish intracellularly, analogous to *S*.Tm$^{ΔinvG}$ and *S*.Tm$^{Δ4}$ (Fig. 3c; MOI-response curves in Supplementary Fig. 4C). Moreover, *S*.Tm$^{ΔsptP}$ was hyperinvasive (Fig. 3c and Supplementary Fig. 4C). This agrees with earlier findings that this effector can counteract SopE/E2 during host cell invasion[46]. Hence, the ability of *S*.Tm strains to trigger a cytokine response in murine BMMCs correlates with their respective ability to invade these cells.

We next asked whether these observations generalize to human MCs, therefore turning to the nontransformed MC line LUVA[47]. LUVA cells did not secrete IL-6 (<16 pg/ml), but rather appreciable amounts of TNF (-700 pg/ml) upon *S*.Tm$^{wt}$ infection (Supplementary Fig. 4D). Most importantly, both this cytokine response, and the capacity of *S*.Tm to invade LUVA cells, was dramatically attenuated by SopB/E/E2 deletion (Supplementary Fig. 4D, E), in full agreement with the data above.

As the *S*.Tm$^{ΔsopBEE2}$ strain failed to promote invasion and cytokine secretion in murine and human MCs, the individual roles of SopB, SopE, and SopE2 were explored further. Deletion of only SopE or SopE2 in isolation, or SopE/E2 in combination, had no effect on *S*.Tm's capacity to elicit IL-6 secretion from BMMCs (Fig. 3d). However, single deletion of SopB resulted in a drop in IL-6 secretion (Fig. 3d). The number of BMMCs harboring intracellular (*ssaG*-GFP + ) *S*.Tm was also modestly (-twofold) reduced by SopB deletion (Fig. 3e; MOI-response curves in Supplementary Fig. 4F). This hints towards a role for SopB in stimulating MC cytokine secretion, and a less prominent role in driving MC invasion. Also, in PCMCs, the *S*.Tm$^{ΔsopB}$ strain elicited lower IL-6 protein secretion and *Il6* transcript levels at 4 hp.i. (Fig. 3f, g). It should, however, be noted that in BMMC cultures from another wild-type mouse strain (C57BL/6J; Jackson), *S*.Tm$^{ΔsopBEE2}$ again triggered negligible IL-6 secretion, while we could not substantiate a non-redundant role for SopB (Supplementary Fig. 4G). Taken together, these data demonstrate that the TTSS-1 effectors SopB/E/E2, working in partial redundancy with each other, promote a swift and full-blown MC cytokine secretion response to *S*.Tm infection.

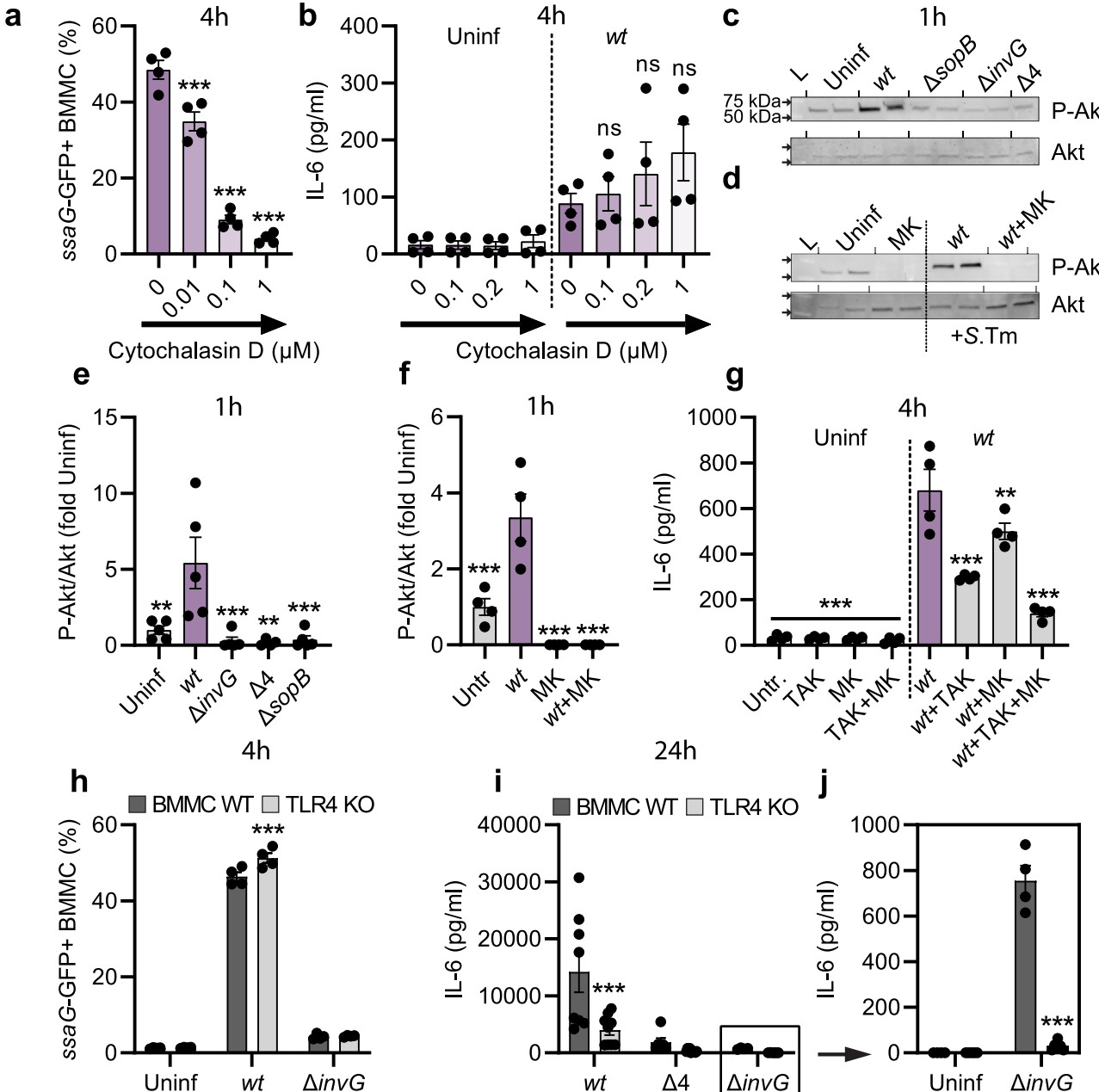

**Fig. 5 | TLR4 and Sop effectors drive cytokine secretion from *Salmonella*-invaded mast cells. a** BMMCs were pretreated for 1 h with the Cyto D concentrations indicated, and infected with MOI 50 of *S*.Tm$^{wt}$ SL1344 carrying the *ssaG*-GFP reporter. After 4 h, BMMCs harboring vacuolar *S*.Tm were quantified ($n = 4$). **b** Similar setup as in (**a**), but IL-6 secretion was measured ($n = 4$). **c** BMMCs were left uninfected or infected with MOI 50 of *S*.Tm$^{wt}$ SL1344 or the indicated TTSS-mutants. After 1 h, cells were harvested and analyzed by immunoblot for P-Akt and Akt. **d** BMMCs were pretreated with MK-2206 for 1 h, infected with MOI 50 of *S*.Tm$^{wt}$, and analyzed as in (**c**). L = ladder. **e**, **f** Quantification of C-D ($n = 5$ for (**e**), except "Δ4" where $n = 4$, $n = 4$ for (**f**)). **g** BMMCs were pretreated with TAK-242 and/or MK-2206 for 30–45 min and infected with MOI 50 of *S*.Tm$^{wt}$. After 4 h, IL-6 secretion was measured ($n = 4$). **h**, **i** BMMCs from TLR4 KO mice and corresponding

WT BMMCs were left uninfected, or infected with MOI 50 of *S*.Tm$^{wt}$ SL1344 or the indicated TTSS-mutants. After 4 h, BMMCs harboring vacuolar *S*.Tm were quantified ($n = 4$) (**h**), and after 24 h IL-6 secretion was measured (BMMC WT $n = 4$ for "Δ*invG*", $n = 6$ for "Δ4", $n = 8$ for others; TLR4 KO n = 10 for "Δ4", $n = 6$ for "Δ*invG*", $n = 9$ for others) (**i**, **j**). **j** depicts an enlargement of (**i**) with independent statistical analysis (two-way ANOVA and Sidak's posthoc test for both). Experiments were performed 2–3 times and mean ± SEM of pooled biological replicates is shown. Data was statistically analyzed with one-way ANOVA and Dunnett's posthoc test, using *S*.Tm$^{wt}$-infected cells for comparisons to all other groups (**a**, **b**, **e–g**). **h**, **i** TLR4 KO BMMCs were compared with corresponding WT BMMC groups by two-way ANOVA with Sidak's posthoc test. **P < 0.01; ***P < 0.001; ns nonsignificant. Exact P values given in source data.

## *Salmonella*-invaded mast cells comprise the source of cytokine production

Based on the results above, it appeared likely that *S*.Tm-invaded MCs were specifically responsible for cytokine production. However, crosstalk between invaded host cells and non-infected bystanders, which respond by cytokine secretion, has been described in other contexts[21,48,49], and could not be ruled out. To distinguish between

these possibilities, we turned to GFP-based flow cytometry sorting of BMMCs, following infection with *S*.Tm$^{wt}$/*ssaG*-GFP (Fig. 4a). The *ssaG*-GFP + BMMC fraction expressed elevated *Il6* transcript levels, similar to the total unsorted population, and significantly higher than the *ssaG*-GFP- BMMC fraction (Fig. 4a). The difference between *ssaG*-GFP+ and *ssaG*-GFP- fractions was, however, relatively modest (Fig. 4a). This might be explained by either that (1) bystander BMMCs also express

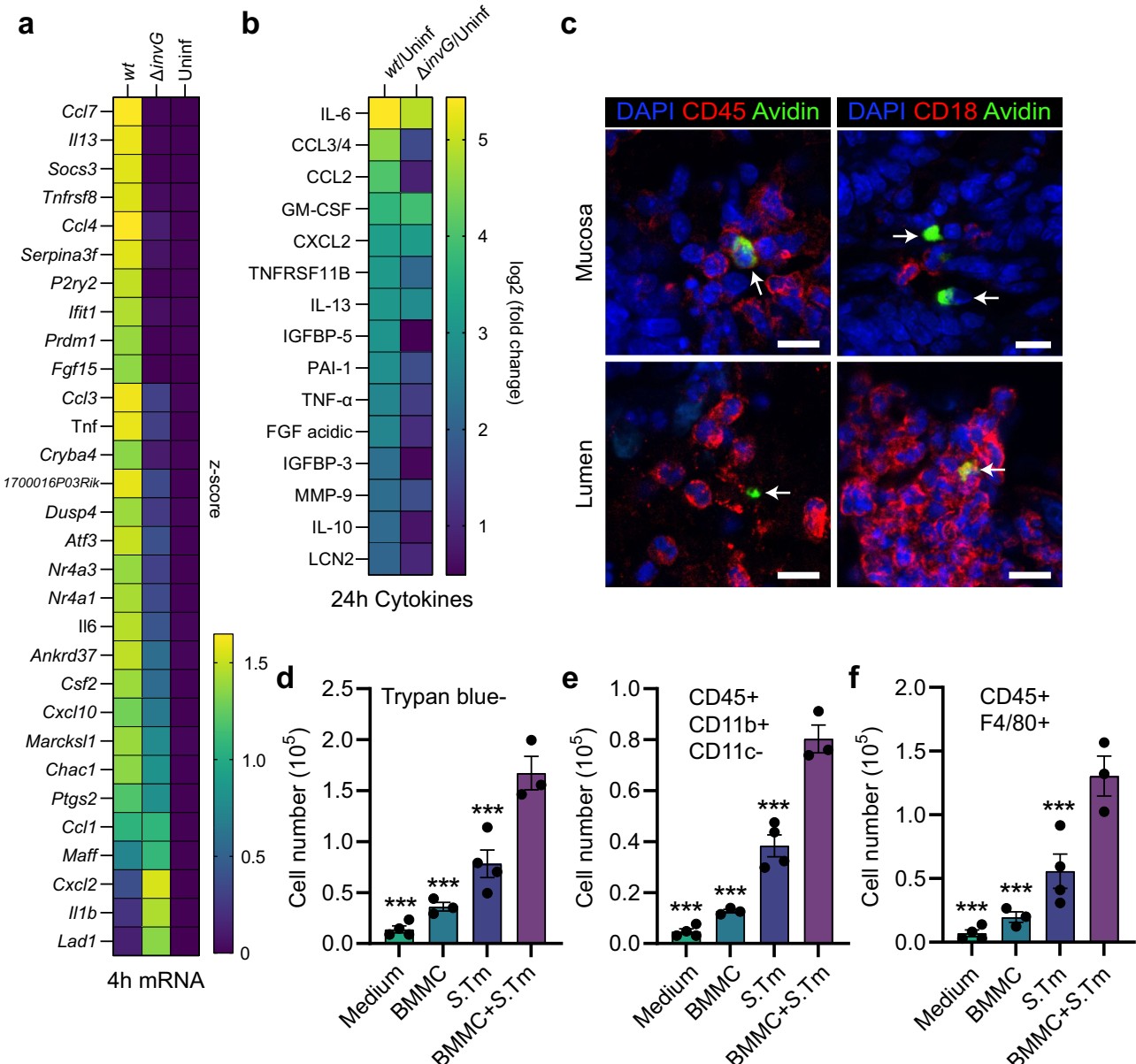

**Fig. 6 | *Salmonella* induces a broad transcriptional and cytokine secretion response in mast cells with functional consequences on myeloid cells. a** RNA sequencing of BMMCs left uninfected, or infected with MOI 50 of *S.*Tm^wt^ or *S.*Tm^ΔinvG^ SL1344 for 4 h, presented as the top 30 significantly upregulated genes between *S.*Tm^wt^-infected and uninfected control, displayed as a z-score-transformed heatmap. Genes are sorted by the formula "*S.*Tm^wt^−*S.*Tm^ΔinvG^" to highlight differences between those two groups. **b** Heatmap of relative log2-fold changes between the indicated groups, derived from a cytokine array of 24-h supernatants from BMMCs infected with MOI 50 of *S.*Tm^wt^ or *S.*Tm^ΔinvG^ SL1344. **c** Representative IF images of MCs in close contact to CD45+ and CD18+ immune cells in the *S.*Tm^wt^ SL1344 -infected intestinal mucosa and lumen at 48 h p.i. Arrows indicate MCs, scale bars: 10 μm. **d** Trypan blue-based live cell counts of bone marrow nucleated cells, cultured for 7 days in base medium supplemented with either medium alone, 24 h uninfected BMMC supernatant, *S.*Tm inoculum-conditioned supernatant, or supernatant of BMMCs infected with *S.*Tm^wt^ for 24 h. **e, f** Similar setup as in (**d**), but quantification of the total number of CD45⁺ CD11b⁺ CD11c⁻ cells (containing monocytes) (**e**) and CD45⁺ F4/80⁺ cells (containing macrophages) (**f**) in bone marrow cultures treated with the indicated supernatants, or base medium alone. **d–f** Data were statistically analyzed with one-way ANOVA and Dunnett's post hoc test, using "BMMC + *S.*Tm" for comparisons to all other groups. Experiments were performed two times and mean ± SEM of pooled biological replicates is show, $n = 4$ ($n = 3$ for BMMCs and BMMCs + *S.*Tm), derived from bone marrow cultures of individual mice. ***$P < 0.001$. Exact $P$ values given in source data.

appreciable levels of *Il6*, or that (2) the *ssaG*-GFP⁻ fraction contains some invaded BMMCs where the *S.*Tm had not turned on the reporter. Staining infected BMMCs for *Salmonella* LPS revealed option 2 to be true (Fig. 4b and Supplementary Fig. 5A–D; further supported by comparisons in Supplementary Fig. 2B).

To separate invaded and bystander BMMCs by more definite means, we conducted infections in transwell plates with filters of 0.4 μm pore size. BMMCs were added to both top and bottom well compartments, but *S.*Tm^wt^ was only inoculated in one of the compartments (either top or bottom; Fig. 4c). Strikingly, BMMCs in the *S.*Tm-inoculated compartment responded with vigorous production of *Il6*, *Il13* and *Tnf* transcripts (Fig. 4d–f). In the non-infected compartment, BMMCs can still be exposed to diffusible bacterial PAMPs, secreted BMMC factors, or DAMPs released from dying MCs, but this only led to marginally elevated cytokine transcription (Fig. 4d–f). Transcript levels were in fact ~7.5-fold (*Tnf*), ~12-fold (*Il6*), to even ~400-fold (*Il13*) higher in the *S.*Tm-infected than in the bystander compartment (Fig. 4d–f). Hence, invaded MCs are the

main cytokine responders upon infection with TTSS-1-proficient invasive *S*.Tm.

## Combined TLR4 and SopBEE2 signals fuel cytokine secretion from *Salmonella*-infected mast cells

*S*.Tm express TLR ligands, most notably the TLR4 agonist LPS and the TLR5 agonist flagellin, which could contribute to MC activation. Measurable levels of TLR4 were detected on BMMCs (Supplementary Fig. 6A). Accordingly, BMMCs responded with IL-6 secretion upon stimulation with pure *E. coli* LPS, but not flagellin (Supplementary Fig. 6B). The LPS response was blocked by preincubating with the TLR4 inhibitor TAK-242 (Supplementary Fig. 6B). Notably, the levels of IL-6 secretion elicited by LPS alone was still >25-fold lower than for a *S*.Tm$^{wt}$ infection (compare Supplementary Fig. 6B and Fig. 3a). As shown in Supplementary Fig. 6C, TAK-242 pre-incubation also attenuated the IL-6 response of BMMCs to *S*.Tm$^{wt}$ infection by ~40%, without significantly affecting the number of BMMC-associated *S*.Tm (Supplementary Fig. 6D). Hence, TLR4 activation contributes to, but is on its own insufficient to account for the total cytokine secretion response of *S*.Tm-invaded MCs. This makes sense considering that both invasive and noninvasive S.Tm strains carry LPS, while only the former elicits full-blown MC cytokine secretion (Figs. 2 and 3).

To test the relationship between *S*.Tm invasion, the TTSS-1 effectors SopB, SopE, and SopE2, and MC cytokine secretion, we next analyzed the effect of blocking bacterial internalization. Cytochalasin D (Cyt D) treatment prior to infection decreased the number of *S*.Tm$^{wt}$ invasion events (i.e., % *ssaG*-GFP+ BMMCs) in a dose-dependent manner (Fig. 5a). Still, these BMMCs secreted IL-6 to a similar extent as in the absence of Cyt D pretreatment (Fig. 5b). This suggests that while TTSS-1 and SopB/E/E2 fuel full-blown cytokine secretion (Figs. 2 and 3), internalization of *S*.Tm into the MCs is not strictly required.

Translocated SopB has in other cell types been shown to trigger the PI3-kinase-Akt pathway[50–52], which can be a potent pro-inflammatory signal. In line with this, *S*.Tm$^{wt}$ elicited highly elevated levels of phosphorylated Akt (P-Akt) in BMMCs, in sharp contrast to *S*.Tm$^{ΔsopB}$, *S*.Tm$^{Δ4}$ and *S*.Tm$^{ΔinvG}$ strains that all fail to express or translocate SopB (Fig. 5c, e). Elevated P-Akt levels were also observed in Cyt D-pretreated *S*.Tm$^{wt}$-infected BMMCs (Supplementary Fig. 6E), again uncoupling this response from bacterial internalization. Both baseline P-Akt levels, and the elevated levels of P-Akt induced by *S*.Tm$^{wt}$ could be abrogated by the selective Akt inhibitor MK-2206 (Fig. 5d, f). Pretreatment with this inhibitor also notably reduced IL-6 secretion from *S*.Tm$^{wt}$-infected BMMCs (Supplementary Fig. 6F). Since lower levels of IL-6 were still detectable under this condition (Supplementary Fig. 6F), we hypothesized that TLR4-dependent (downstream of LPS sensing) and Akt-dependent (downstream of the *S*.Tm TTSS-1 effectors) signals may combine to elicit cytokine secretion. The experimental data supported this notion; pretreatment with TAK-242 + MK-2206 caused more pronounced reduction of IL-6 secretion from *S*.Tm$^{wt}$-infected BMMCs than either inhibitor alone (Fig. 5g).

From these data, we postulated a two-step activation mechanism whereby extracellular *S*.Tm can be sensed by TLR4, generating a weak MC cytokine response. Upon *S*.Tm invasion, this TLR4 signal combines with signals elicited by the TTSS-1 effectors SopB/E/E2 (including SopB-triggered Akt phosphorylation), leading to swift and full-blown MC cytokine secretion. To formally test this model, we established BMMCs from *Tlr4*$^{-/-}$ (C57BL/6J) mice. As expected, *S*.Tm invaded WT and *Tlr4*$^{-/-}$ BMMCs with similar proficiency and in a TTSS-1-dependent manner (Fig. 5h). At 4 h p.i., *S*.Tm$^{wt}$ and *S*.Tm single-effector mutants could still elicit IL-6 secretion also from *Tlr4*$^{-/-}$ BMMCs, but *S*.Tm$^{ΔinvG}$, *S*.Tm$^{Δ4}$ and *S*.Tm$^{ΔsopBEE2}$ did not (Supplementary Fig. 6G). When extending the time frame (analysis at 24 h p.i.), *Tlr4*$^{-/-}$ BMMCs were found to produce ~2–3-fold lower

levels of IL-6 than WT BMMCs in response to *S*.Tm$^{wt}$ (Fig. 5i). Strikingly, the noninvasive *S*.Tm strains essentially failed to elicit a response altogether in the absence of TLR4 (Fig. 5i, j). We conclude that MCs can use a two-step activation mechanism and tune their cytokine secretion output to differentiate between a noninvasive vs. invasive *S*.Tm encounter.

## The broad-scale cytokine response from *Salmonella*-infected mast cells can promote myeloid cell survival and differentiation

To define the global MC gene expression changes elicited by invasive and noninvasive *S*.Tm, we next performed RNA Sequencing (RNASeq) of BMMCs left uninfected, exposed to *S*.Tm$^{wt}$, or to *S*.Tm$^{ΔinvG}$ (MOI 50; analysis at 4 h p.i.). Principal component analysis (PCA; based on all transcripts) illustrated a clear separation of the three sample groups along the PC1 axis (Supplementary Fig. 7A). Pairwise comparisons of either *S*.Tm$^{wt}$-infected vs. uninfected MCs, or *S*.Tm$^{wt}$- vs *S*.Tm$^{ΔinvG}$-infected MCs, showed that invasive infection brought about pronounced upregulation of a large panel of transcripts (Supplementary Fig. 7A–F; 3471 transcripts significantly upregulated between *S*.Tm$^{wt}$-infected and uninfected samples). Among these were mRNAs encoding IL-6, TNF, and IL-13, hence validating our results by qPCR and ELISA (Figs. 2–4). Additional cytokine/chemokine transcripts induced by invasive infection were e.g., *Ccl3*, *Ccl4*, *Ccl7*, *Cxcl10*, and *Csf2* (encoding GM-CSF) (Fig. 6a). Another set of transcripts exemplified by *Il1b* showed equal levels of induction by *S*.Tm$^{wt}$ and *S*.Tm$^{ΔinvG}$ (Fig. 6a), again corroborating the earlier qPCR results (Supplementary Fig. 3E, K). Moreover, quantification of select transcripts (informed by Fig. 6a) in infected LUVA cell showed that this differential transcript response to *S*.Tm$^{wt}$ vs *S*.Tm$^{ΔinvG}$ was generalizable also to human MCs (Supplementary Fig. 7G).

In a subsequent approach, a semi-quantitative array detecting 111 cytokines was used to screen supernatants of BMMCs either left uninfected, exposed to *S*.Tm$^{wt}$, or to *S*.Tm$^{ΔinvG}$ (MOI 50; analysis at 24 h p.i.) (Fig. 6b). 15 cytokines showed a >fourfold increase between *S*.Tm$^{wt}$-infected and uninfected BMMCs, while 5 were also >fourfold increased between *S*.Tm$^{ΔinvG}$-infected and uninfected samples (Fig. 6b). The highest fold change between *S*.Tm$^{wt}$-infected and uninfected BMMCs was seen for IL-6, followed by CCL3, CCL2, GM-CSF, and then by IL-13 and TNF, in full agreement with our previous ELISA analyses (Supplementary Fig. 2C, D, M, N). Overall, this cytokine profile can be interpreted as a mixed pro-inflammatory (e.g., IL-6, TNF) and immunomodulatory (e.g., IL-10, PAI-1) output that may foster recruitment of granulocytes (e.g., CCL3, CXCL2), dendritic cells and monocytes (e.g., CCL2, CCL3), as well as promote survival and differentiation of macrophages (e.g., GM-CSF). Indeed, when further scrutinizing gut tissue harvested from *S*.Tm-infected mice, mucosal and luminal MCs were found to be surrounded by high densities of CD45 (broad hematopoietic cell marker) and CD18 (marker for monocytes, macrophages, and granulocytes) positive cells (Fig. 6c).

In light of these observations, we finally examined the functional capacity of infected MC secretions, by culturing nucleated mouse bone marrow cells for 7 days in a base medium (see "Methods") that on its own failed to support cell survival (Fig. 6d). The base medium was mixed with 24 h conditioned supernatants from either uninfected BMMCs, the *S*.Tm$^{wt}$ inoculum alone, or *S*.Tm$^{wt}$-infected BMMCs. Supernatants from naive BMMCs had minimal impact on the bone marrow cells, but *S*.Tm-conditioned supernatants enhanced cell survival and macrophage differentiation to some extent (Fig. 6d–f; flow cytometry gating shown in Supplementary Fig. 8). However, both of these stimulatory effects were drastically higher for supernatants harvested from *S*.Tm$^{wt}$-infected BMMCs. We conclude that MC secretions elicited by *S*.Tm infection enhance survival of bone marrow-derived progenitors and may act in concert with soluble *S*.Tm components to promote myeloid cell differentiation.

## Discussion

This work establishes that MCs, common innate immune cells of mucosal tissues, can tune their cytokine response to extracellular vs. invasive forms of the prototype enterobacterium *S*.Tm and close relatives. Prior studies have shown that MCs respond to PAMPs of both gram-negative and -positive bacteria through TLRs[6–9]. We corroborate these findings, showing that *E. coli* LPS, or noninvasive *S*.Tm strains, trigger a TLR4-dependent cytokine response. Importantly, however, this weak response, observed also upon exposure to noninvasive *E. coli* and *Y. pseudotuberculosis* strains, vastly undershoots the maximal capacity of MCs to initiate de novo transcription and cytokine production. Instead, a full-blown MC response requires a second activation step directly linked to TTSS-1-dependent *S*.Tm invasion effectors. Through this wiring, MCs appear capable of informing their surrounding of a bacterium's virulence potential. Experiments separating *S*.Tm-infected from neighboring MCs localized the vigorous MC response specifically to the bacterium-invaded cell population. This excludes that the second MC activation step is elicited by DAMP release from cells damaged by the infection (further supported by von Beek et al.[16]). Moreover, an *S*.Tm strain that retains TTSS-1 translocon function in the absence of TTSS-1 effectors (*S*.Tm$^{\Delta 4}$) did not recapitulate the MC response to *S*.Tm$^{wt}$. This also excludes that the second activation step comprises sensing of plasma membrane perturbation, as has been noted for MCs exposed to bacterial cytolysins[14–16]. Rather, *S*.Tm single- and multiple gene mutant infections, and pharmacological inhibition assays, point to the translocated TTSS-1 effectors SopB, SopE, and SopE2 as responsible for the invasion-linked signal(s). These effectors have previously been linked to pro-inflammatory transcription in epithelial cells[50]. SopB alters plasma membrane phosphatidylinositol phosphate (PIP) pools to generate e.g., PI(3,4)P$_2$, through phosphatase and phosphotransferase activities[53,54]. This recruits multiple kinases and promotes phosphorylation/activation of Akt, with consequences[50–52,55] on host cell transcription and cell survival. We found that MCs exposed to *S*.Tm$^{wt}$ exhibited elevated levels of phospho-Akt, while this was not observed for strains incapable of SopB translocation (*S*.Tm$^{\Delta invG}$, *S*.Tm$^{\Delta 4}$, *S*.Tm$^{\Delta sopB}$). MC pretreatment with the Akt inhibitor MK-2206 also attenuated the MC cytokine response. Notably, however, the magnitude of the SopB contribution varied between MC models, whereas a *S*.Tm$^{\Delta sopBEE2}$ triple mutant consistently elicited a similar low response as the TTSS-1-deficient strains. SopE/E2 can, in partial redundancy with SopB, activate Rho GTPases such as Rac1 and Cdc42, which promote *S*.Tm invasion, but may also elicit pro-inflammatory transcription through MAP kinase and/or Nod1 pathways[50,56,57]. It therefore seems most likely that invasive *S*.Tm spark a mixed transcription-stimulating signal in MCs through translocated SopB/E/E2, triggering both Akt activation, and additional pathways redundantly activated by the three effectors.

Our study also weighs in on the question if MCs permit bacterial internalization. Previous reports suggested that enterobacteria like *S*.Tm are not efficiently taken up by MCs, and thereby fuel a more restricted MC response than, for example, *Staphylococcus aureus*[6,21]. However, we demonstrate here that *S*.Tm$^{wt}$ grown under TTSS-1-inducing conditions efficiently invade and establish an intracellular niche (evident by *ssaG*-GFP expression and TEM analyses) within both mouse (BMMC, PCMC) and human (LUVA) MC models. By contrast, genetically TTSS-1-deficient bacteria (*S*.Tm$^{\Delta invG}$), or *S*.Tm$^{wt}$ grown under non-TTSS-1-inducing conditions (overnight culture in LB with vigorous shaking) had an essentially noninvasive phenotype in cultured MCs. The discrepancy between our and previous findings are likely explained by how the *S*.Tm inoculum was prepared. In either case, our results favor that (i) MCs have a minimal inherent capacity for phagocytosis of enterobacteria, but that (ii) they are highly susceptible to TTSS-1-mediated active *S*.Tm invasion. It is here noteworthy that SopB-dependent Akt phosphorylation has been shown to promote host cell survival and *S*.Tm expansion in the intracellular niche, e.g., within B

cells[58]. It appears plausible that *S*.Tm in a similar way could benefit from invading MCs as a niche for long-term persistence and/or growth. The intracellular enterobacterial lifestyle(s) within MCs remain an intriguing topic for further research.

According to the two-step activation mechanism proposed here, MCs are capable of a graded response to bacterial infection that depends on the pathogen's invasive capacity. Notably, this response does not include overt degranulation, as evident from both β-hexosaminidase release and CD63 staining assays. IgE-mediated activation of FcεRI receptors elicits prompt MC degranulation[59], but this is typically not seen upon bacterial detection, although some reports have linked degranulation to certain microbial stimuli[60,61]. Our data do not exclude that the *S*.Tm TTSS-1 effector SptP can have a MC degranulation-suppressing effect "in trans", that is when degranulation is stimulated by other means, as has been proposed by others[40]. However, neither *S*.Tm$^{wt}$, nor *S*.Tm$^{\Delta sptP}$, elicited above-background degranulation of cultured MCs, suggesting that degranulation plays a minimal role in the immediate MC response to this bacterium. Instead, extracellular *S*.Tm, *E. coli*, or *Y. pseudotuberculosis*, or pure *E. coli* LPS sensed through TLR4, gave rise to a weak production of cytokines and chemokines. Upon two-step detection of invasive *S*.Tm through both TLR4- and SopBEE2-elicited pathways, this cytokine/chemokine response was both faster and more vigorous. While not the main focus of this study, we also detected type I interferons in response to *S*.Tm$^{wt}$, which is in line with earlier proposals linking intracellular localization of a pathogen to boosted type I interferon production[6]. Among the secreted proteins strongly stimulated by invasive *S*.Tm were both typical pro-inflammatory (e.g., IL-6, TNF) and immunomodulatory (e.g., IL-10) cytokines, chemokines linked to granulocyte (e.g., CCL3, CXCL2) and monocyte (e.g., CCL2, CCL3) recruitment, and growth factors (e.g., FGF, GM-CSF). Indeed, we could also substantiate that MCs in the *S*.Tm-infected gut frequently colocalize with a variety of other immune cell types, and that supernatants from *S*.Tm$^{wt}$-infected MCs boosted immune cell progenitor survival, as well as myeloid cell differentiation.

How can this MC response be integrated into the current understanding of enterobacterial infection in vivo? In the mouse experiments, MCs were found in the intestinal submucosa/mucosa already prior to infection, and the mucosal MC population expanded by ~twofold in the infected group. We also found MCs transmigrating into the *S*.Tm-filled lumen, coming in direct contact with the bacteria in the process. It is well established that *S*.Tm express TTSS-1 in the lumen and use this to invade epithelial cells and elicit acute inflammation through epithelial inflammasome activation[29,62]. During transition across the mucosa, TTSS-1 expression is downregulated[63], generating a stealthier *S*.Tm phenotype in deeper tissues. Hence, it appears most plausible that the MC response to TTSS-1-proficient *S*.Tm is predominantly relevant in the bacterium-laden superficial mucosa and lumen. Our combined data from fixed mucosal tissue microscopy, MC culture infections, and bone marrow co-cultures, suggest that the secretory output produced upon one-step or two-step MC activation will promote optimal shaping of the local immune cell microenvironment at these infection sites. The triggering of inflammation, immune cell influx and maturation represents a double-edged sword during enterobacterial infection. In the case of *S*.Tm, the acute host response can restrict pathogen translocation across the gut mucosa, but when dysregulated also disrupt the epithelial barrier and promote pathogen overgrowth[64,65]. MCs have in other gram-negative infection models been shown to drive neutrophil influx and bacterial clearance[66]. At the same time, MC-derived TNF has also been found to worsen disease outcome and fuel bacterial colonization upon intraperitoneal *S*.Tm infection[67]. Hence, it seems critical for MCs (as well as other mucosal cell types) to carefully adjust their inflammation-modulatory output to the properties of the intruder and thereby foster a protective, rather than deleterious, overall counter-response. This study provides

proof-of-principle evidence that MCs can indeed grade their cytokine output by combining classical PAMP sensing with effector-triggered immunity[68,69], which enables them to differentiate between non-invasive and invasive enterobacterial infection.

## Methods

### Mice for mast cell culture

For experiments involving $Tlr4^{-/-}$ BMMCs, B6(Cg)-$Tlr4^{tm1.2Karp}$/J (#029015) and corresponding C57BL/6J WT controls (#000664), 8 weeks old female mice were purchased from The Jackson Laboratory. For all other experiments involving bone marrow or BMMCs, C57BL/6 WT mice (8–14 weeks old; male and female), bred and maintained at the National Veterinary Institute (SVA, Uppsala, Sweden) were used. The experimental procedures were approved by the local animal ethics committee (Uppsala djurförsöksetiska nämnd, Dnr 5.8.18-05357/2018). Whenever possible, remaining bones from WT C57BL/6 mice used as controls in other experiments were acquired for culturing BMMCs.

### Mouse infections

Eight-week-old female CBA mice (Charles River) were acclimatized to the new environment for one week before infection. 24 hours before infection, mice were pretreated with 20 mg streptomycin per oral gavage. Mice were deprived of food and water for 4 h prior to infection by oral gavage with $3.0–7.5 \times 10^6$ CFU S.Tm in 100 µl Dulbecco's phosphate-buffered saline (PBS). For infection, S.Tm were grown in LB overnight, diluted 1:20 and re-grown for 4 h before diluting bacteria in PBS for infection. The infection dose was confirmed by viable counts on streptomycin plates. Mice were monitored frequently for signs of unhealth. The experiments were approved by the local animal ethics committee (Umeå djurförsöksetiska nämnd, Dnr A27-17) and mice were housed in accordance with the Swedish National Board for Laboratory Animals guidelines.

### Cryosectioning of cecum, immunofluorescence, and toluidine staining of paraffin sections

For preservation in paraffin, tissues were fixed in 4% PFA for 6–12 h at room temperature, rinsed in PBS and 70% ethanol, and immediately dehydrated and paraffinized in Tissue-Tek®VIP (SAKURA). Tissues were embedded with HistowaxTM paraffin (Histolab) with Tissue-Tek®TEC (SAKURA) and kept at 4 °C until sectioning. For cryo-embedding, tissue pieces were placed in 4% PFA/ 4% sucrose/PBS solution for 6–12 h at room temperature, followed by incubation in 20% sucrose/ PBS (overnight 4 °C). Excess liquid was removed, tissue placed in OCT, flash frozen in liquid $N_2$ and stored at −80 °C. Cryo-sections of 20 µm were cut on a Cryostat CryoStar NX70 (Epredia) with at least 40 µm distance between sections, placed on Superfrost Plus Adhesion Microscope Slides (Thermo Fisher Scientific, #J1800AMNT) and dried >16 h. Sections were rehydrated in PBS (Gibco, #70013-016 or #14190144) for 5 min and permeabilized for 3 min in PBS/0.1% Triton X-100. Slides were stained with 2.5 µg/ml DAPI (Sigma-Aldrich, #D9542), 4U/ml phalloidin-A647 (Thermo Fisher Scientific, #A22287) and 10 µg/ml avidin-A488 (Thermo Fisher Scientific, #A23170) for 40 min. After washing 3× in PBS for 3 min, mounting was done with Mowiol 4-88 (Sigma-Aldrich, #81381), and slides were dried overnight before storing at 4 °C. MCs were counted as avidin+ cells per cecum section. For staining of S.Tm, slides were blocked in 10% normal goat serum (Sigma-Aldrich, #G9023) in PBS for 30 min after the permeabilization step, followed by 40 min incubation with Salmonella O Antiserum Factor 5, Group B 1:250 in blocking buffer. After washing, slides were stained as described above but including Goat-α-rabbit-IgG(H + L)-Cy3 (Molecular probes, #A10520) 1:200 in PBS. For staining with CD18 and CD45 (both 1:50), 1:200 Goat-α-Rat-IgG(H + L)-AF647 (Invitrogen, #10666503) was used without phalloidin instead. For staining of Salmonella, all washing steps were performed only for 1 min to avoid loss of bacteria. For toluidine-stained tissue sections, 5 µm

paraffin tissue sections were cut in Microm HM360 (Zeiss), placed on SuperFrost slides and dried (50 °C for one hour or overnight at 37 °C). Sections were deparaffinized in xylene at 60 °C (3 × 10 min) and rehy-drated through a graded series of alcohol. Sections were incubated with 0.1% toluidine in 1% NaCl pH 2.3–2.5, water, 95% ethanol, and mounted in dibutylphtalate xylene (DPX) (Sigma-Aldrich) after dehy-dration in 99% ethanol and xylene. All primary antibodies and their dilutions used in this study are shown in Supplementary Table 4.

### Bone marrow-derived mast cell culture

Bone marrow from tibiae and femurae of one mouse per culture was flushed out with PBS, washed 1× at $300\times g$ for 7 min and filtered through a 70-µm cell strainer. Cells were resuspended in 50 ml BMMC culture medium consisting of 90% DMEM (Fisher Scientific, #31966047), 10% heat-inactivated FBS (Thermo Fisher Scientific, #D5671) and supplemented with Penicillin–Streptomycin (100 U/ml, 100 µg/ml, Sigma-Aldrich, #P0781) and 10 ng/ml recombinant IL-3 (Peprotech, #213-13). Medium was changed every 3–4 days to a fresh flask in the first 4 weeks of culture, maintaining a cell density of $0.5 \times 10^6$ cells/ml. Afterward, medium was changed every 3–7 days. BMMCs were used during 4–10 weeks of culture. $Tlr4^{-/-}$ BMMCs were used up to 4 months of culture. Cell numbers were determined by trypan blue (Thermo Fisher Scientific, #15250-061) exclusion and quantified by an automated cell counter (CountessTMII FL, Life Technologies).

### Mouse bone marrow cell culture

Mouse bone marrow was extracted as described above until the cell strainer filtering step. Erythrocytes were lysed by resuspending the pellet in 3 ml of RBC Lysis Buffer (eBioscience, #00-4333-57) and incubating for 3 min on ice. Nucleated bone marrow cells were washed in PBS and resuspended to $10^6$ cells/ml in BMMC culture medium. 900 µl conditioned medium was placed in a 12-well plate and 100 µl of bone marrow cells were added. After 7 days, cells were harvested with cell lifters (Corning, #3008) and stained for flow cytometry as described below.

### Peritoneal cell-derived mast cell culture

After euthanizing the mouse, the abdomen was washed with 70% ethanol. The abdominal cavity was carefully cut open and the abdo-men was rinsed with 5 ml ice-cold PBS by injecting and gently shaking the mouse. Lavage from each mouse was collected individually to avoid possible contamination. After collection, the lavage was cen-trifuged for 8 min, $300\times g$ and resuspended in PCMC culture medium (BMMC culture medium with 50 µM 2-mercaptoethanol (Sigma-Aldrich, #M6250), 20 ng/ml SCF (Peprotech, #250-03) and IL-3. On the next day, cells were observed under the microscope and cells from separate mice were pooled into one culture. Medium was changed every 3–4 days and the cell density was adjusted to $0.5 \times 10^6$ cells/ml.

### LUVA cell culture

LUVA cells ([47]; Kerafast, #EG1701-FP) are derived from untransformed CD34+ enriched mononuclear hematopoietic progenitor cells, obtained from a blood donor and cultured in the presence of IL-3, IL-6, and SCF. The cells were maintained in complete StemPro™-34 SFM (Thermo Fisher Scientific, #10639011), supplemented with 2 mM L-glutamine or 1× GlutaMAX™ (Thermo Fisher Scientific, #35050061) and penicillin–streptomycin and subcultured every 3–4 days.

### Salmonella strains, plasmids, and culture conditions

All strains and plasmids used in this study can be found in Supple-mentary Tables 1 and 2, respectively. In Fig. 2 and Supplementary Fig. 2, wild-type and mutant strains of Salmonella enterica serovar Typhi-murium ATCC 14028 background were included[70]. All other strains used in this study were of a SL1344 background (SB300; streptomycin

resistant)[71]. The S.Tm$^{\Delta sptP}$ mutant was generated via transfer of a previously described deletion[70] from a S.Tm 14028 strain (C1172) to the SL1344 background by P22 transduction. For infections, S.Tm cultures were grown overnight at 37 °C for 12 h in LB 0.3 M NaCl with appropriate antibiotics on a rotating wheel incubator to optimize aeration, followed by subculturing in the same medium without antibiotics at a 1:20 dilution for 4 h at 37 °C. For non-virulence-inducing conditions, S.Tm was grown for 20 h at 37 °C in LB under 190 rpm shaking. Prior to infection, 1 ml was spun down for 4 min at 12,000×$g$ and reconstituted in co-cultivation medium (cell-specific standard culture medium without IL-3 and antibiotics). E. coli strains were grown overnight in LB at 37 °C and subcultured in LB for 2 h at 37 °C prior infection. Y. pseudotuberculosis were grown overnight in LB at 26 °C. Bacteria were subcultured for 1 h at 26 °C in LB, containing 50 mM of CaCl₂, followed by 1 h shifting to 37 °C. When using different bacterial species, ODs were adjusted to equal bacterial concentrations, based on CFU numbers. After infection, inocula were diluted 1:10⁶ and 50 µl were plated onto agar plates with antibiotics when appropriate to enumerate CFUs.

### Infection of mast cells with S.Tm

For all infections, mast cells (BMMCs, PCMCs or LUVA) were washed twice in PBS and resuspended in the respective culture medium without antibiotics (co-cultivation medium). In experiments performed for ELISA or RT-qPCR analysis, if not indicated otherwise, 500 µl of 1 × 10⁶ mast cells/ml were added to 24-well plates (Sarstedt) and infected with 25 µl inoculum, resulting in a multiplicity of infection (MOI) of 50, for 30 min in 37 °C, 5% CO₂. Afterward, gentamicin was added to a final concentration of 90 µg/ml, leaving the intracellular bacteria intact. After an additional 3.5 h, wells were harvested in tubes, centrifuged for 5 min at 400×$g$ and supernatants and pellets were frozen separately. For degranulation assays, the incubation after the addition of gentamicin was reduced to 30 min and phenol red-free DMEM (Gibco, #31053028) was used in the co-cultivation medium. To generate samples for immunoblots, 1–2 ml of cells were infected in 12- or 6-well plates with identical concentrations of gentamicin and bacteria but only 30 min incubation after the addition of gentamicin and prior to freezing, the pellets were washed with PBS. For experiments involving flow cytometry, 180 µl of 0.556 × 10⁶ mast cells/ml were added to 96-well round bottom plates (Thermo Fisher Scientific, #163320) and infected by adding 20 µl of bacteria to the indicated MOIs. After 30 min of incubation as above, plates were gently centrifuged for 3 min, 200×$g$, supernatants were discarded, and the plates vortexed gently before adding 200 µl/well of co-cultivation medium, containing 100 µg/ml gentamicin. Plates were incubated for further 3.5 h, washed 1× in 1% BSA (Sigma-Aldrich # A9418) in PBS (200×$g$, 3 min), and fixed in 2% PFA (Sigma-Aldrich, #158127) in the dark for 20-30 min. After 1× washing, cells were resuspended in 1% BSA in PBS and stored at 4 °C until flow cytometry analysis of mCherry or GFP-positive MCs. MK-2206 (Selleck chemicals, #S1078), TAK-242 (Sigma-Aldrich, #614316) or Cyt D (Sigma-Aldrich #C8273) were added diluted in medium 30–45 min prior to infection. E. coli LPS (Sigma-Aldrich, #L4516) and flagellin (Sigma-Aldrich #SRP8029) were added diluted in medium and used as indicated in the respective figure legend. Transwell plates were from Sigma-Aldrich (#CLS3470).

### Microscopy

For fluorescence microscopy, 400 µl of fixed BMMCs (0.5 × 10⁶ cells/ml) in 1% BSA in PBS were added to 24-well glass bottom plates (High-performance #1.5 cover glass, Cellvis, #P24-1.5 P). Images were acquired on a custom-built microscope, based on a Nikon Eclipse Ti2 body, in differential interference contrast (DIC) and green fluorescence channels with a Prime 95B (Photometrics) camera through a 100X/1.45 NA Plan Apochromatic objective (Nikon), using a X-light V2 L-FOV spinning disk (Crest Optics, Italy) and a Spectra-X-light engine (Lumencor). Micro-Manager was used for controlling the microscope (µManager plugin[72]).

For TEM, BMMCs infected with S.Tm MOI 50 for 4 h (gentamicin present after 30 min), or left uninfected, were fixed in 2.5% glutaraldehyde (Ted Pella) + 1% paraformaldehyde (Merck) in PIPES pH 7.4 and stored at 4 °C until further processed. Samples were rinsed with 0.1 M PB for 10 min prior to 1 h incubation in 1% osmium tetroxide (TAAB) in 0.1 M PB. After rinsing in 0.1 M PB, samples were dehydrated using increasing concentrations of ethanol (50%, 70%, 95%, and 99.9%) for 10 min at each step, followed by 5 min incubation in propylene oxide (TAAB). The samples were then placed in a mixture of Epon Resin (Ted Pella) and propylene oxide (1:1) for 1 h, followed by 100% resin and left overnight. Subsequently, samples were embedded in capsules in newly prepared Epon resin and left for 1–2 h and then polymerized at 60 °C for 48 h. Ultrathin sections (60–70 nm) were cut in an EM UC7 Ultramicrotome (Leica) and placed on a grid. The sections were subsequently contrasted with 5% uranyl acetate and Reynold's lead citrate and visualized with Tecnai™ G2 Spirit BioTwin transmission electron microscope (Thermo Fisher/FEI) at 80 kV with an ORIUS SC200 CCD camera and Gatan Digital Micrograph software (both from Gatan Inc.). Cecal tissue cryosections stained for fluorescence microscopy were imaged using a LSM700 (Zeiss) confocal microscope equipped with a Plan-Apochromat 40x/ 0.95 Korr M27 objective with pinhole set to 1 AU for each wavelength, at the BioVis platform of Uppsala University. Fiji[73] was used for image analysis and distance measurements.

### Supernatant analyses by ELISA and β-hexosaminidase assays

BMMC and PCMC supernatants were analyzed by ELISA for IL-6, TNF, and IL-13 (Invitrogen, # 88-7064-76, #88-7324-88, #88-7137-88). LUVA supernatants were assayed for IL-6 and TNF (Invitrogen, #88-7066-88, #88-7346-88). β-hexosaminidase was measured in fresh supernatants by transferring 20 µl sample/well to a 96-well plate (Sarstedt) and adding 80 µl of 1 mM 4-nitrophenyl N-acetyl-β-D-glucosaminide (Sigma-Aldrich, #N9376) in citrate buffer with pH 4.5. After 1 h of incubation at 37 °C, the reaction was stopped with 200 µl Na₂CO₃ buffer at pH 10 and absorption was measured at 405 nm. Total β-hexosaminidase content was acquired by lysis with 1% Triton X-100 (Sigma-Aldrich, T8787) prior to the assay and for the respective infection experiment. MCs treated with 2 µM calcium ionophore A23187 (Sigma-Aldrich, #C7522), were included as positive control for β-hexosaminidase release. Phenol red-free DMEM was used as medium for measuring β-hexosaminidase.

### Real-time quantitative PCR

Total RNA was isolated from frozen pellets and cecum tissue using the NucleoSpin RNA extraction kit (Machery-Nagel, #740955.5) or the RNeasy Plus Mini Kit (Qiagen, #74134). Equal concentrations of RNA (measured by NanoDrop and 1 µg for the cecum tissue) were reverse transcribed to cDNA using the iScript™ cDNA Synthesis Kit (Bio-Rad, #1708891) and stored at −20 °C. cDNA was diluted 1:5 (for cecum 1:1 or 1:10) and qPCR was performed on a CFX384 Touch™ (Bio-Rad) with iTaq Universal SYBR (Bio-Rad, #1725121) using 1 µl of cDNA and 200 nM of reverse and forward primers (sequences are listed in Supplementary Table 3). PCR was performed according to the manufacturer's instructions. If not indicated otherwise, data is shown as fold change (2$^{-\Delta\Delta Cq}$) to uninfected MCs, normalized to Hprt transcription. Statistics were calculated based on ΔCq values.

### Flow cytometry

For detection of TLR4, 1 × 10⁶ BMMCs were resuspended in 2% BSA (Sigma-Aldrich) in PBS, containing 0.5 µg/ml of Fc-block anti-CD16/32 (1:1000, BD Life Sciences, #553142) and incubated for 5 min at RT. The cells (100 µl each) were either left unstained, or stained with 5 µg/ml (1:100) anti-CD284 (TLR4) antibody (clone UT41) conjugated with Alexa Fluor 488 (Invitrogen, #53-9041-82) or identical amount of mouse IgG1 kappa isotype control (Invitrogen, #MG120). After 30 min incubation on ice, the cells were washed and resuspended in 2% BSA in

PBS. For CD63 cell surface staining, BMMCs were resuspended in 1% BSA in PBS, containing 0.5 μg/ml of Fc-block anti-CD16/32 and incubated for 5 min at RT. Cells were stained for 30 min on ice, washed and resuspended in 1% BSA in PBS. For LPS staining, fixed cells were resuspended in 100 μl of 0.1% saponin (VWR, Calbiochem, #558255), 1% BSA and 1:250 of *Salmonella* antiserum in PBS and incubated for 1 h at RT in the dark. After washing in 0.1% saponin, 1% BSA in PBS, the cells were incubated for 30 min with 1:200 Goat-α-rabbit-IgG(H + L)-Cy3 in the dark and washed in 0.1% saponin again. Flow cytometry was performed using a MACSQuant VYB (Miltenyi biotec). For bone marrow culture analysis, cells were collected in 2% FBS in PBS containing fluorophore-conjugated antibodies targeting the following surface markers: CD45, CD11b, CD11c, F4/80 (all 1:100). After 30 min of incubation, the cells were washed twice with 2% FBS in PBS, and resuspended in the same buffer for analysis. Flow cytometry was performed on a Cytoflex LX (Beckman Coulter). Data analysis was performed using FlowJo (BD Biosciences) version 10.8.1. Detailed information of all primary antibodies can be found in Supplementary Table 4.

### Cell sorting
After infection of BMMCs as described above for 4 h, cells were washed in 1% BSA in PBS and resuspended in cold MACSQuant® Tyto® Running Buffer (Miltenyi Biotech, #130-107-207) to a concentration of $0.5 \times 10^6$ cells/ml. All further steps were performed at 4 °C. Cells were sorted on *ssaG*-GFP- expression in a MACS® GMP Tyto® Cartridge (Miltenyi Biotech #170-076-011) until purity of GFP- cells reached 99%. This procedure isolated GFP- BMMCs and enriched for GFP+ cells. After sorting, cells were washed in 1% BSA in PBS, pelleted and frozen until RNA extraction.

### Protein extraction and immunoblotting
Frozen pellets were resuspended in cold lysis buffer consisting of RIPA buffer (ThermoScientific, #89900) with protease inhibitors (Roche, #4693132001) and PhosSTOP™ (Roche, #4906845001), using 75 μl per $1 \times 10^6$ cells. After 30 min of incubation on ice, samples were centrifuged for 20 min at 4 °C max. speed to remove debris. Supernatants were transferred in fresh tubes, measured with the bicinchoninic acid assay (Pierce, #23227) and frozen until further use. Equal concentrations of protein (10–25 μg) were mixed with 4x Laemmli buffer (Bio-Rad, #1610747) containing 5% 2-mercaptoethanol, boiled for 5 min at 95 °C and loaded together with 5 μl of ladder (Bio-Rad, #1610373) on 4–20% Mini-PROTEAN® TGX Stain-Free™ Protein Gels (Bio-Rad, #4568094 or #4568096). Gels were run for 25 min at 200 V and transferred to nitrocellulose (Bio-Rad, #1704158) with a Trans-Blot Turbo Transfer System (Bio-Rad), using the program for intermediate molecular weight. 1 min activation of the gel and imaging of total protein on the blot was performed by a Gel Doc EZ Imager (Bio-Rad). Blots were blocked in Intercept blocking buffer (PBS, Li-cor, #927-70001) for 1 h at RT, incubated 2 h on RT or overnight at 4 °C with antibodies for Akt (1:1000, Cell Signaling, #9272) or P-Akt (Phospho-Akt (Ser473) (D9E) XP® Rabbit mAb (1:2000, Cell signaling, #4060) diluted in blocking buffer. After 3x washing in PBS-Tween (0.05%), blots were incubated with Goat-α-rabbit-IgG-HRP (1:10,000, Cell Signaling, #7074) for 1 h at RT. After washing, blots were incubated with ECL Prime (Cytiva, #GERPN2232) for 5 min at RT under shaking and imaged with a ChemiDoc MP (Bio-Rad). Two blots were run in parallel for Akt and P-Akt, antibody signal was normalized to total protein on the membrane, respectively, and data were displayed as P-Akt of total Akt, normalized to untreated cells.

### RNA sequencing
Sequencing libraries were prepared from 500 ng total RNA using the TruSeq stranded mRNA library preparation kit (Illumina Inc., #20020595,) including polyA selection. Unique dual indexes (Illumina Inc., 20022371) were used. The library preparation was performed according to the manufacturer's protocol (#1000000040498). The quality of the libraries was evaluated using the Fragment Analyzer from Advanced Analytical Technologies, Inc. using the DNF-910 dsDNA kit. The adapter-ligated fragments were quantified by qPCR using the Library quantification kit for Illumina (KAPA Biosystems) on a CFX384 Touch instrument (Bio-Rad) prior to cluster generation and sequencing. Library preparation and sequencing was performed by the SNP&SEQ Technology Platform, a national unit within the National Genomics Infrastructure (NGI), hosted by the Science for Life Laboratory in Uppsala, Sweden (scilifelab.se/units/ngiuppsala). Sequencing was carried out on an Illumina NovaSeq 6000 instrument (NSCS v 1.7.5/RTA v 3.4.4) according to the manufacturer's instructions. Demultiplexing and conversion to FASTQ format were performed using the bcl2fastq2 (2.20.0.422) software, provided by Illumina. Additional statistics on sequencing quality were compiled with an in-house script from the FASTQ-files, RTA and BCL2FASTQ2 output files. The RNA-seq data were analyzed using the best practice pipeline nf-core/rnaseq. Detailed information about the analysis pipeline can be found here: ngisweden.scilifelab.se/bioinformatics/rna-seq-analysis and nf-co.re/sarek. For differential expression analysis, DESeq2 1.38.3 in combination with R 4.2.2 and RStudio 2022.12.0 + 353 were used. Transcriptome data can be accessed at Gene Expression Omnibus under GSE223601.

### Cytokine array
Proteome Profiler Mouse XL Cytokine Array (Bio-Techne, #ARY028) was used, following the manufacturer's instruction for fluorescent detection with IRDye® 800CW Streptavidin (LI-COR, #926-32230) on an Odyssey CLx Infrared Imager. Raw images were cropped, converted to 16 bit with ImageJ and dot intensities were analyzed with Image Lab 6.1 (Bio-RAD). Ratios of log2-fold changes between groups were calculated.

### Statistical analysis
If not indicated otherwise, all graphs were plotted with Prism 9.5.1 (GraphPad), and statistical analysis was performed either with one-way analysis of variance (ANOVA) and Dunnett's posthoc test or two-way ANOVA and Sidak's posthoc test in order to compare groups to either control or wt-infected MCs as indicated in the figure legends. Whenever appropriate, paired or unpaired *t* tests or for data using mice, the Mann–Whitney *U* test was used. Significance levels were: $*P < 0.05$, $**P < 0.01$ and $***P < 0.001$. If not indicated otherwise, for every n, the mean of all MC wells infected with an individual subculture derived from an individual overnight culture serves as a single data point. If not indicated otherwise, every experiment was performed at least twice on different days.

### Reporting summary
Further information on research design is available in the Nature Portfolio Reporting Summary linked to this article.

## Data availability
The RNA-seq data generated in this study have been deposited in the GEO under accession number GSE223601. The rest of the data are available in the article, Supplementary Information, or Source Data file. Source data are provided with this paper.

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

## Acknowledgements

We thank members of the Sellin and Pejler laboratories for helpful discussions. We are grateful for support regarding TEM and fluorescence microscopy from BioVis, Uppsala University. Flow cytometry was also in part performed on instruments provided by the BioVis facility. This work was supported by the Knut and Alice Wallenberg Foundation (KAW 2016.0063 to M.F. and M.E.S.), the Swedish Research Council (2016-00803 to J.H., 2020-00882 to G.P., and 2018-02223 to M.E.S.), a Lennart Philipson Award (MOLPS, 2018 to M.E.S.), and the SciLifeLab Fellows program.

## Author contributions

Conceptualization: C.v.B., G.P. and M.E.S.; methodology: C.v.b., A.F., P.G., M.L.D.M., E.M.-E. and J.E.; investigation: C.v.b., A.F., O.L., G.I.P., and E.M.-E.; formal analysis: C.v.b., A.F. and E.M.-E.; interpretation: C.v.b., A.F., P.G., M.L.D.M., O.L., E.M.-E., J.H., M.F., G.P., and M.E.S.; resources: J.H., M.F., G.P., and M.E.S.; supervision: J.H., M.F., G.P. and M.E.S.; funding acquisition: J.H., M.F., G.P. and M.E.S.; visualization: C.v.B. and M.L.D.M.; writing—original draft: C.v.B. and M.E.S.; writing—reviewing and editing: all authors.

## Funding

## Competing interests

The authors declare no competing interests.
