## [Peer Review File · Nature Communications]

A Two-Step Activation Mechanism Enables Mast Cells to Differentiate their Response between Extracellular and Invasive Enterobacterial InfectionREVIEWER COMMENTS

Reviewer #1 (Remarks to the Author):

The authors present a well-structured manuscript on the activation capacity of invasive enteropathogens in mast cells and propose a very interesting mechanism of distinct activation pattern by invasive versus non-invasive pathogens. By using elegant tools, the authors provide evidence for a two-step mechanism by which a full-blown activation of mucosal mast cells is triggered by invasive but not altered (non-invasive) germs.

Although the *in vitro* work demonstrates convincing evidence of such a two-step activating mechanism, the reader inevitably comes to the question whether such mechanism has a relevance and consequence for the *in vivo* course of invasive infections. In principle, the authors have all necessary tools available, i.e. TLR-4 KO mice and several altered strains of *Salmonella Typhimurium* (S.Tm), to complement their studies by also demonstrating the role and relevance of the proposed mechanism *in vivo*. In Figure 1 of their manuscript, the authors nicely show *in vivo* results of mice infected by WT S.Tm. Thus, it would be very interesting to additionally demonstrate whether infection of TLR-4 KO versus WT mice by differently altered S-Tm strains could translate the *in vitro* data into the *in vivo* situation, e.g. by demonstrating the intensity of invasion and the inflammatory response (infiltration by macrophages, etc.), which the authors show preliminary by studying the effects of mast cell supernatants infected by S.Tm strains on murine bone marrow cultures.

Reviewer #2 (Remarks to the Author):

Summary: Authors carefully and convincingly show that mast cells have a differential cytokine response to extracellular vs. invasive strains of *Salmonella Typhimurium* (S. Tm) *in vitro*. Challenge of mast cells with more invasive bacterial strains resulted in more robust cytokine secretion, which could result in increased immune cell recruitment, survival, and function.

Major Comments:

1-It is well known that S. Tm requires intestinal inflammation to sustain its replication in the intestinal tract. Therefore, does the increased IL-6 release by mast cells result in greater clearance of the bacteria by recruited cells? Or does the mast-cell response lead to greater inflammation that results in increased resources for bacterial replication? In other words, can the authors speculate on how interactions between mast cells and invasive strains of S.Tm. influence the outcome of the infection? Studies published elsewhere may illustrate on this point (PMID: 20035049) and should be discussed here.

2-The images of infected mast cells from Figure 2A suggest that granules are empty or depleted at the 4h post-infection mark, but no B-hexosaminidase was detected extracellularly. One possibility is that the granules are being mobilized to the phagosome. If this is the case, is the viability of the internalized S. Tm bacteria affected by this process (how long does GFP Signal stay after death of bacterium)?

3-What is the fate of mast cells that have internalized bacteria at later time points (24h or later)? If the cell is lysing, they could be releasing DAMPs that are known to also induce IL-6 production and could be magnifying the IL-6 loop by activating non-infected mast cells.

4- In relation to comment 4, it would be nice to see a quantification of the distance/interaction implied in figure 1 C-E. Each figure has only a couple of mast cells, and not all of them are close to bacteria. Additionally, the authors note regarding figure 1J that the avidin+ staining did not always colocalize with DAPI, suggesting they are dying off in the lumen. This could be further evidence of a mast cell death pathway and IL-6 production.

5-Fig. 4b: The authors suggest that the ssaG-GFP-fraction may contain some invaded BMMCs where the *S. Tm* had not turned on the reporter. Then, is it possible the authors are underestimating the % of infected BMMCs shown in Figs. 3C or Fig. 3e?

6-There is a concern about the validity of findings shown in Fig. 3F-G (and potentially other figures) considering that BMMCs generated from B6/J do not recapitulate the phenotype for SopB.

Minor Comments:

1-Please indicate whether *S. Tm* (ATCC 14028) is a TTSS-1 proficient Salmonella.

2-It is unclear from sentence in lines 147-148 whether the authors challenge the view that mast cells sense bacteria TLR applies only to Salmonella or to other bacteria as well. This needs further clarification.

3-Please indicate what the gene Nr4a3 is encoding for and its significance here.

4-The readers would greatly benefit from a table that outlines each bacterial strain and their functional relevance.

Uppsala

June 27th, 2023

Dear colleagues,

Thank you for the constructive feedback on our manuscript, NCOMMS-23-10790, entitled “A Two-Step Activation Mechanism Enables Mast Cells to Differentiate their Response between Extracellular and Invasive Enterobacterial Infection”. In response to the reviewer input, we have now conducted several additional experiments. These include i) extension of the microscopy-based analysis of mast cells and their relationship to bacteria and other neighboring cell types in the *Salmonella*-infected mouse gut, ii) confirmation that predictions from our model hold true also for other enterobacterial species/strains (*Escherichia coli* and *Yersinia pseudotuberculosis*), and iii) further control experiments instilled by the reviewer critique. Moreover, we have made appropriate clarifications throughout the manuscript text. Below we give our response to the specific comments of the expert reviewers and detail how they have been considered in the revised manuscript. Reviewer comments are inserted as italicized text, followed by our point-by-point reply.

Reviewer #1

The authors present a well-structured manuscript on the activation capacity of invasive enteropathogens in mast cells and propose a very interesting mechanism of distinct activation pattern by invasive versus non-invasive pathogens. By using elegant tools, the authors provide evidence for a two-step mechanism by which a full-blown activation of mucosal mast cells is triggered by invasive but not altered (non-invasive) germs.

Response: We are pleased to hear of this positive assessment by reviewer 1.

*Although the in vitro work demonstrates convincing evidence of such a two-step activating mechanism, the reader inevitably comes to the question whether such mechanism has a relevance and consequence for the in vivo course of invasive infections. In principle, the authors have all necessary tools available, i.e. TLR-4 KO mice and several altered strains of *Salmonella Typhimurium* (S.Tm), to complement their studies by also demonstrating the role and relevance of the proposed mechanism in vivo. In Figure 1 of their manuscript, the authors nicely show in vivo results of mice infected by WT S.Tm. Thus, it would be very interesting to additionally demonstrate whether infection of TLR-4 KO versus WT mice by differently altered S-Tm strains could translate the in vitro data into the in vivo situation, e.g. by demonstrating the intensity of invasion and the inflammatory response (infiltration by macrophages, etc.), which the authors show preliminary by studying the effects of mast cell supernatants infected by S.Tm strains on murine bone marrow cultures.*

Response: This point is well taken and we have ourselves dwelled extensively on ways to further probe the consequences of S.Tm – mast cell interactions *in vivo*. However, studying the progression of infection and inflammatory cell influx in WT and TLR4 KO mice (with various S.Tm mutants) will unfortunately not be informative. TLR4 is also expressed by

many other mucosal cell types, and some of us have in a recent study shown that “sentinel macrophages” located in the lamina propria respond potently to *S.Tm* infection through TLR4 and within minutes secrete inflammatory mediators (¹). To untangle contributions specifically of TLR4-dependent versus -independent mast cell (MC) mechanisms during *in vivo* infection would at a minimum require mucosal MC-specific TLR4 KO mice, which are currently unavailable. Moreover, even if such mice would be available, we and others have established that also, e.g., epithelial cell inflammasome signaling potently contributes to triggering of acute gut inflammation and the first wave of inflammatory cell influx into the murine gut (²⁻⁴). Hence, multiple cell types (at least intestinal epithelial cells, lamina propria sentinel macrophages, MCs, and possibly others) will in parallel contribute to tuning the inflammatory state of the *S.Tm*-infected gut. This highly redundant wiring of mucosal inflammatory circuits, combined with the lack of mouse models for selective and clean ablation of specific mucosal MC signaling modules, prohibits a deeper global analysis of the MC effects during *in vivo* infection.

Nevertheless, from the available data it appears most plausible that activated MCs contribute to local effects in their direct vicinity during mature stages of *S.Tm* gut infection (when the invasive bacteria and MCs can come in contact with reasonable frequency; see Fig 1 and results presented in the new Fig S1). To gain insights about the local environment around MCs in the inflamed gut, we have therefore instead expanded our confocal microscopy analyses of tissue sections from *S.Tm*-infected mice. When co-staining for MCs (avidin) and broad immune cell markers (e.g. CD45 and CD18), we find MCs to be surrounded by dense aggregations of such CD45+/- CD18+ cells, a phenomenon observed both around tissue-lodged MCs, and around MC remnants located in the inflamed gut lumen. Fluorescence micrographs that illustrate these findings are presented in the new Fig 6C and described in the corresponding results text. These *in vivo* observations substantiate that MCs in the *S.Tm*-infected gut come in close proximity to a range of blood-derived infiltrating cells, hence supporting that the stimulatory effects of infected MC secretions seen in the bone-marrow co-culture experiments (presented in Fig 6C-E of the original manuscript, now Fig 6D-F) should be possible also *in vivo*. We have elaborated on this point also in the revised discussion (line 372 onwards).

In addition to this extension of the *in vivo* experimentation, we have also examined if our explanatory model holds true across other enterobacterial species, beyond the *Salmonella* SL1344 and 14028 strains and the pure *E.coli* LPS tested initially. If our two-step model for MC activation is broadly applicable, one would expect that non-invasive *E.coli*, and *Yersinia pseudotuberculosis* (that similar to *Salmonella* has a type-three-secretion system, but uses it to prevent uptake into host cells rather than to promote it) should elicit similar weak MC activation as observed for the genetically attenuated non-invasive *S.Tm* strains (“ $\Delta invG$ ” and “ $\Delta 4$ ”). Indeed, in these new experiments, cultured MCs produced modest and similar levels of IL-6 when exposed for 24h to either *E.coli* DH10B, *E.coli* MG1655, *Yersinia pseudotuberculosis*, or *S.Tm* ^{$\Delta invG$} , while again the invasive wildtype *S.Tm* elicited a much more vigorous cytokine response. For IL-13, these differences between the four non-invasive enterobacterial strains and the invasive *S.Tm* were even more striking. These new data, presented in Fig 2J and Fig S2J of the revised manuscript provides further support for that the two-step model of MC activation by enterobacteria is broadly generalizable. We hope that these extensions will satisfy the concern of reviewer 1 regarding the general implications of our findings.

Reviewer #2

Summary: Authors carefully and convincingly show that mast cells have a differential cytokine response to extracellular vs. invasive strains of Salmonella Typhimurium (S. Tm) in vitro. Challenge of mast cells with more invasive bacterial strains resulted in more robust cytokine secretion, which could result in increased immune cell recruitment, survival, and function.

Response: We are happy to learn that reviewer 2 finds the presented evidence compelling, and we also appreciate the constructive criticism. Below we have addressed the remaining concerns of this reviewer.

Major Comments:

1-It is well known that S. Tm requires intestinal inflammation to sustain its replication in the intestinal tract. Therefore, does the increased IL-6 release by mast cells result in greater clearance of the bacteria by recruited cells? Or does the mast-cell response lead to greater inflammation that results in increased resources for bacterial replication? In other words, can the authors speculate on how interactions between mast cells and invasive strains of S.Tm. influence the outcome of the infection? Studies published elsewhere may illustrate on this point (PMID: 20035049) and should be discussed here.

Response: Indeed, the inflammatory output from any host cell type activated during bacterial infection can have either host-protective, or infection-promoting effects, and sometimes even both depending on the context. The Piliponsky et al 2010 study mentioned by reviewer 2 is a perfect example for this, since it illustrates that MCs can promote mouse survival in a cecal ligation and puncture model of modest peritonitis, but that at the same time MC-derived TNF worsened the outcome of severe peritoneal *Salmonella* infection. This is a relevant topic also for our present study; thank you for pointing this out.

As detailed in response to reviewer 1, it is not possible to fully disentangle the impact of TLR4-dependent and -independent MC secretions in the *S.Tm*-infected gut *in vivo*, due to i) the lack of mice selectively ablated for signaling components only in mucosal MCs, ii) the rapid inflammation kinetics of the murine infection model (goes from uninflamed to fully inflamed within a few hours), and iii) the highly parallel wiring of inflammatory circuits in the murine gut (⁵). Still, we have been able to better characterize the cellular microenvironment around MCs in the infected gut. This microscopy analysis provides important context for the function-oriented *ex vivo* experiments presented in the last figure (see updated Fig 6C-F). Moreover, in response to this reviewer comment, we have extended the discussion on how the mucosal MC response to *Salmonella* may affect the outcome of the infection. This includes discussion of the Piliponski et al 2010 findings, and the possible host-protective, or infection-stimulatory roles of MC mediators (lines 384-393).

Finally, we would like to stress that the main novelty of our study lies in the mapping of a two-step model whereby MCs can respond differently to invasive and non-invasive enterobacteria. This salient point gains support from new results presented in Fig 2J and S2J, which demonstrate that three other non-host-cell-invasive strains of *E.coli/Yersinia* induce similar weak MC responses as observed for the genetically attenuated non-invasive *S.Tm* ^{Δ invG}.

2-The images of infected mast cells from Figure 2A suggest that granules are empty or depleted at the 4h post-infection mark, but no B-hexosaminidase was detected extracellularly. One possibility is that the granules are being mobilized to the phagosome. If this is the case, is the viability of the internalized *S. Tm* bacteria affected by this process (how long does GFP Signal stay after death of bacterium)?

Response: A typical feature of *Salmonella* invasion into host cells is the induction of entry structures that in many types of cells take the form of expansive membrane ruffles. The membrane ruffling response also generates collateral macropinosomes (“Infection-Associated Macropinosomes – “IAMs”;⁶) that later can fuse with the intracellular *Salmonella*-containing vesicle and regulate its size. Since we did not observe degranulation from *S.Tm*-infected MCs, we have considered the seemingly expanded and less dense granules in some infected MCs a result of IAM - granule fusion events. However, the possibility put forward by reviewer 2 here would be an alternative explanation. The intracellular life cycle of *S.Tm* within MCs is a topic we in the future intend to study in much greater depth by real-time and structural microscopy techniques, which may conclusively resolve these possibilities. Nevertheless, to tackle the main point made by the reviewer, we have here conducted time course analyses to see how the *ssaG*-GFP+ MC population varies over time (up to 72h post-infection, with gentamicin added to the medium after the initial 30min infection). The results are presented in Figure R1 here below in the response letter. As evident from these data, the MC population positive for intracellular GFP+ *S.Tm* at 24h remains broadly the same also at 48 and 72h post-infection. This supports the conclusion that once they make it into the MCs, *S.Tm* bacteria can linger therein for considerable time periods. As mentioned, we plan to study these later infection events in much greater depth in follow-up work, and would therefore like to refrain from making declarative statements about the long-term intra-MC life style of *Salmonella* in the present manuscript.

Figure R1. MCs harboring vacuolar *S.Tm*/pssaG-GFP+ persist for up to 72h across a range of MOIs. Quantification of BMMCs harboring vacuolar *S.Tm* (SL1344 background; *pssaG*-GFP reporter). The experiment was performed twice with a total of 4 replicates originating from distinct overnight cultures. Mean \pm SEM of pooled biological replicates are shown. Groups were statistically compared by two-way ANOVA and Dunnett’s posthoc test, comparing 48 and 72h groups to 24h.

3-What is the fate of mast cells that have internalized bacteria at later time points (24h or later)? If the cell is lysing, they could be releasing DAMPs that are known to also induce IL-6 production and could be magnifying the IL-6 loop by activating non-infected mast cells.

Response: In the microscopy and flow cytometry experiments conducted so far, we see only a minor subfraction of MCs succumbing to the infection within the ~1-24h window used. Pronounced MC death was only noted following excessive bacterial exposure (MOI >100 for longer time periods, which may equate to the situation within the *Salmonella*-colonized gut lumen; see example images in Fig 1 and the new Fig S1). Moreover, as evident from Fig R1 above, *S.Tm*-positive MCs can remain within the total MC culture for several days, further illustrating that cell death is not the typical consequence to this infection at modest MOIs.

It still remains possible that DAMPs released from a few lysed MC could modulate the cytokine response of neighboring MCs. However, for the following two reasons we believe this to have a negligible impact on the total cytokine output. 1) In our transwell assays, only the MCs that come in direct contact with the *S.Tm* initiate pro-inflammatory transcription. DAMPs released from dying MCs should reach also the other compartment, but clearly does not initiate a response there (Fig 4C-F). 2) In earlier published work, we directly tested if BMMC lysate can stimulate IL-6 production and release from naïve BMDCs, but could not detect IL-6 secretion above baseline (please see Figure S3 in ⁷). We therefore favor the notion that DAMPs released from *S.Tm*-infected MCs do not have a decisive role in the cytokine response characterized in the present manuscript. This has been spelled out in the revised manuscript text (lines 323-324).

4- In relation to comment 4 [3, sic], it would be nice to see a quantification of the distance/interaction implied in figure 1 C-E. Each figure has only a couple of mast cells, and not all of them are close to bacteria. Additionally, the authors note regarding figure 1J that the avidin+ staining did not always colocalize with DAPI, suggesting they are dying off in the lumen. This could be further evidence of a mast cell death pathway and IL-6 production.

Response: As requested, we have quantified the distance-to-nearest-bacterium among MCs located in the mucosal tissue, or in the gut lumen (rarer) within *S.Tm*-infected regions *in vivo*. These data (new Fig S1) illustrate that luminal MCs/MC remnants co-localize closely with *S.Tm*, whereas the distances between mucosal MC and bacteria are more heterogeneous. We interpret this to mean that encounters between TTSS-1-primed invasive *S.Tm* and MCs should be most prevalent during early stages of mucosal inflammation where both parties occupy the epithelium-proximal space. This has been pointed out in the manuscript text (lines 109-111). As outlined above (see point #3 by this reviewer), we do not believe that DAMPs released from dying MCs contribute to any major extent to the total IL-6 output from MCs.

5-Fig. 4b: The authors suggest that the *ssaG*-GFP-fraction may contain some invaded BMDCs where the *S. Tm* had not turned on the reporter. Then, is it possible the authors are underestimating the % of infected BMDCs shown in Figs. 3C or Fig. 3e?

Response: In response to this reviewer comment, we have experimentally assessed what percentage of total MC-associated *S.Tm* are positive for *ssaG*-GFP reporter expression. These experiments are based on parallel MC infection with otherwise identical *S.Tm*^{wt} strains that harbor either the *pssaG*-GFP reporter, or a constitutive reporter (*prpsM*-mCherry). The new results, presented in Fig S2B, show that across several MOIs relevant to this study, the

ssaG-GFP reporter reveals the large majority of the MC-associated bacteria. Note that the constitutive reporter *S.Tm* strain will also render MCs “positive” when the bacteria merely sit on the cell surface without invading. This explains why the constitutive reporter strain gives a modestly higher percentage of bacterium-positive MCs as compared to the *pssaG*-GFP-carrying one. We conclude that the *pssaG*-GFP reporter informs on the vast majority of the intracellular *S.Tm* within MCs. The results text has been adjusted accordingly.

6-There is a concern about the validity of findings shown in Fig. 3F-G (and potentially other figures) considering that BMMCs generated from B6/J do not recapitulate the phenotype for SopB.

Response: From multiple previous studies of *S.Tm* infections in other contexts, it is evident that the three effectors SopB, SopE, and SopE2 have overlapping and often redundant roles during host cell interaction. For the invasion step itself, SopE and SopE2 directly activate Rho-GTPases to fuel actin nucleation, while SopB indirectly accomplishes the same outcome (Rho-GTPase activation that is) through regulation of plasma membrane inositide pools (^{8,9}). In addition, these three effectors can in a redundant manner boost transcriptional responses in, for example, epithelial cells (¹⁰). Inflammatory transcription-inducing signaling includes both MAPK and AKT pathways that again can operate in partial redundancy with each other. From this, it appears conceivable that minor differences among MC models with respect to either Rho-GTPase composition, or propensity to signal through AKT versus MAPK pathways, may explain the relative impact of specifically the SopB effector. It is here important to note that in all five types of MC cultures tested in our study (1. C57BL/6 BMMCs, 2. C57BL/6J BMMCs, 3. C57BL/6 PCMCs, 4. C57BL/6J *Tlr4*^{-/-} BMMCs, and 5. Human LUVA cells) the *S.Tm* strain lacking all three of the effectors SopB, SopE, and SopE2 (i.e. “*S.Tm*^{Δ*sopBEE2*” consistently elicited vastly lower cytokine transcription and cytokine secretion than the *S.Tm*^{wt} strain, and essentially similar levels as the completely non-invasive “*S.Tm*^{Δ*invG*” strain. The only variation noted in C57BL/6J BMMCs concerns the relative impact of SopB visavi SopE/E2.}}

In response to this reviewer comment, we have now carefully pruned the manuscript text and, where relevant, clarified the redundant nature of signaling downstream of the effectors SopB/E/E2. The conclusions have also been adjusted where relevant to emphasize that “signal number 2” in our two-step MC activation model is produced in a partly redundant manner downstream of these three effectors (e.g. lines 197-199).

Minor Comments:

1-Please indicate whether S. Tm (ATCC 14028) is a TTSS-1 proficient Salmonella.

Response: The *S.Tm* ATCC 14028 strain is TTSS-1 proficient, which we have clarified in the revised manuscript (line 126).

2-It is unclear from sentence in lines 147-148 whether the authors challenge the view that mast cells sense bacteria TLR applies only to Salmonella or to other bacteria as well. This needs further clarification.

Response: We apologize for imprecise wording and have made appropriate clarifications in this text passage. We do not question that TLR signaling is relevant for MCs to recognize both *Salmonella* and other types of bacteria. However, our study provide evidence for a more

elaborate signaling program that allows MCs to produce two levels of cytokine output, a low level when only TLR-signaling is triggered, and a high level when the MCs in addition also detect bacterial invasion. The new data from infections with non-invasive *E.coli* and *Yersinia* strains (presented in Fig 2J and S2J) further support this conclusion.

3-Please indicate what the gene Nr4a3 is encoding for and its significance here.

Response: Nr4a3 (Nuclear receptor 4a3) is a highly expressed transcription factor in BMMCs, involved in cytokine and chemokine production under inflammatory conditions. In earlier work, the *Nr4a3* gene had been found to be the highest upregulated gene in BMMCs infected with *Streptococcus equi* (7,11), which is why we have included it as a useful marker transcript also in this work. The manuscript results text now includes a corresponding clarification (lines 151-152).

4-The readers would greatly benefit from a table that outlines each bacterial strain and their functional relevance.

Response: We have now prepared a table that describes all the bacterial strains used in the study (see Table S1).

We thank the reviewers for the careful scrutiny and constructive input, which we believe has significantly improved the manuscript. In addition to the changes sparked by reviewer comments, we have carefully pruned the manuscript for consistency and wording, which has resulted in a few additional minor adjustments. All changes are highlighted in yellow in the marked-up version of the revised manuscript.

We look forward to your response.

The Authors

References to response letter

1. Hausmann, A. *et al.* Intercrypt sentinel macrophages tune antibacterial NF- κ B responses in gut epithelial cells via TNF. *Journal of Experimental Medicine* **218**, (2021).
2. Sellin, M. E. *et al.* Epithelium-Intrinsic NAIP/NLRC4 Inflammasome Drives Infected Enterocyte Expulsion to Restrict Salmonella Replication in the Intestinal Mucosa. *Cell Host & Microbe* **16**, 237–248 (2014).
3. Rauch, I. *et al.* NAIP-NLRC4 Inflammasomes Coordinate Intestinal Epithelial Cell Expulsion with Eicosanoid and IL-18 Release via Activation of Caspase-1 and -8. *Immunity* **46**, 649–659 (2017).
4. Müller, A. A. *et al.* An NK Cell Perforin Response Elicited via IL-18 Controls Mucosal Inflammation Kinetics during Salmonella Gut Infection. *PLOS Pathogens* **12**, e1005723 (2016).
5. Kaiser, P., Diard, M., Stecher, B. & Hardt, W.-D. The streptomycin mouse model for Salmonella diarrhea: functional analysis of the microbiota, the pathogen's virulence factors, and the host's mucosal immune response. *Immunological Reviews* **245**, 56–83 (2012).
6. Stévenin, V. *et al.* Dynamic Growth and Shrinkage of the Salmonella-Containing Vacuole Determines the Intracellular Pathogen Niche. *Cell Reports* **29**, 3958-3973.e7 (2019).
7. von Beek, C. *et al.* Streptococcal sagA activates a proinflammatory response in mast cells by a sublytic mechanism. *Cell. Microbiol.* e13064 (2019).
8. Hume, P. J., Singh, V., Davidson, A. C. & Koronakis, V. Swiss Army Pathogen: The Salmonella Entry Toolkit. *Frontiers in Cellular and Infection Microbiology* **7**, 348 (2017).
9. Fattinger, S. A., Sellin, M. E. & Hardt, W.-D. Salmonella effector driven invasion of the gut epithelium: breaking in and setting the house on fire. *Current Opinion in Microbiology* **64**, 9–18 (2021).
10. Bruno, V. M. *et al.* Salmonella Typhimurium Type III Secretion Effectors Stimulate Innate Immune Responses in Cultured Epithelial Cells. *PLOS Pathogens* **5**, e1000538 (2009).
11. Rönnberg, E., Guss, B. & Pejler, G. Infection of Mast Cells with Live Streptococci Causes a Toll-Like Receptor 2- and Cell-Cell Contact-Dependent Cytokine and Chemokine Response. *Infect Immun* **78**, 854–864 (2010).

REVIEWER COMMENTS

Reviewer #1 (Remarks to the Author):

I do thank the authors for their extensive response and considerations related to the comments given by the reviewers. From my point of view the all comments were intensively addressed, discussed, and where indicated translated into an improved outline and wording the manuscript. I have no further comments.

Reviewer #2 (Remarks to the Author):

No additional comments. Thank you for the thorough responses to our comments.

Reviewer #3 (Remarks to the Author):

In the manuscript by von Beek et al., the authors present a well-structured manuscript, which is logically presented and easy to follow. The authors show that mast cells have distinct cytokine responses to extracellular and invasive *Salmonella enterica* Typhimurium strains in vitro and that mast cells migrate to close proximity of the infecting pathogen in the cecum in vivo. This revised manuscript has improved after revision but there are still some questions that remain unanswered.

Comments and concerns:

1. The authors challenge Nramp^{+/+} CBA mice and look at acute responses, what is the rationale of looking at mast cells during this acute time point. Would the authors expect similar responses during a more chronic time point e.g. post 10 days of infection?

2. The authors discuss their findings in relation to previous findings saying that mast cells are found in the intestinal submucosa and mucosa prior to infection and that the population expands after infection. Except for expanding and being invaded, what do the authors propose that the infiltrating mast cells do in vivo?

In vitro the authors convincingly show that invading STm trigger the release of TNF. Based on previous studies of mast cells in the urinary tract, the Abraham lab have shown that mast cell derived TNF is essential for infiltrating neutrophil activation, do the authors propose a similar mechanism taking place in the cecum of STm infected mice? The same lab has also shown that mast cell derived IL-10 is essential for controlling bacterial burden, could this be a reason for the mast cells expansion in the mucosa in vivo?

Furthermore, this reviewer has concerns that the only read out of the mast cells in vivo is their number and their proximity to bacteria. What is the activation state of the mast cells? Do they have increased CD63 expression to signify degranulation? These questions could easily be addressed by comparing neutrophil responses by flow cytometry or looking at epithelial integrity by FITC-dextran in WT vs mast cell deficient Kit^{W-sh/W-sh} mice during acute and more chronic infection stages. In addition, if the authors could show by flow cytometry that luminal and/or mucosal mast cells have STm inside would greatly support the in vitro findings of the paper and help distinguish whether these cells show signs of "slow" extracellular TLR4 activation or if this is due to intracellular activation.

3. One of the strengths of the paper is that the authors translate findings from BMMCs and

PCMCs to the human mast cell line and show some similar phenotypes between these systems. The paper would however improve further by validating some of the RNA-seq hits in the human in vitro system, and thereby showing that these findings are translatable between mouse and human cells.

4. The authors show that mast cells do not degranulate after 1h while they are transcriptionally active at this time point. Does this mean that all of the released cytokines synthesized de novo during the time of infection? The majority of the assays take place at 4 or 24 hours post infection, how do the β -hex levels compare at these time points? Furthermore, is intracellular STm a necessity for degranulation? in that case investigating the kinetics of the uptake and correlate this to degranulation would support the authors point.

5. The authors convincingly show that STm invade mast cells, however, there is limited discussion to why STm would want to invade these cells. To this reviewer's knowledge, mast cells do not act as antigen presenting cells, nor migrate from inflamed tissue to peripheral sites, so how do Salmonella benefit from invading these cells? Investigating STm replication capability in mast cells using the gent-protection assay and plating CFUs or by microscopy/flow cytometry using the pFCcgi plasmid system would answer this question and greatly improve the knowledge of STm/mast cell interactions.

The authors further show that sopB activate AKT phosphorylation and discuss how this can affect downstream cytokine activation. In line with the previous comment, sopB has also been shown to phosphorylate AKT in B cells in a way to promote Salmonella survival and this should perhaps be discussed as a potential effect of the phenotype observed for sopB infected mast cells and not only that it may drive cytokine production.

6. Since the authors eloquently show that there are impaired cytokine expression and release in MCs infected by the different mutants, do the authors believe that there would be a phenotype in mice as well? e.g. would there be less mucosal/luminal MCs than in SL1344WT infected mice?

Minor comments:

The flow plots in Supplementary figure 5 are not properly compensated

Uppsala

November 7th, 2023

Dear colleagues,

Thank you for the assessment and comments on our revised manuscript, NCOMMS-23-10790R, entitled “A Two-Step Activation Mechanism Enables Mast Cells to Differentiate their Response between Extracellular and Invasive Enterobacterial Infection”. Below we give our response to the reviewer comments, which are inserted verbatim as italicized text, followed by our point-by-point reply. The corresponding text changes and descriptions of new data in the manuscript have been indicated therein by yellow highlight.

Reviewer #1

I do thank the authors for their extensive response and considerations related to the comments given by the reviewers. From my point of view the all comments were intensively addressed, discussed, and where indicated translated into an improved outline and wording the manuscript. I have no further comments.

Reviewer #2

No additional comments. Thank you for the thorough responses to our comments.

Response: We thank reviewer #1 and #2 for the constructive input on the original version of the manuscript. We are happy to learn that both reviewers find our revisions thorough and complete.

Reviewer #3

*In the manuscript by von Beek et al., the authors present a well-structured manuscript, which is logically presented and easy to follow. The authors show that mast cells have distinct cytokine responses to extracellular and invasive *Salmonella enterica* Typhimurium strains *in vitro* and that mast cells migrate to close proximity of the infecting pathogen in the cecum *in vivo*. This revised manuscript has improved after revision but there are still some questions that remain unanswered.*

Response: We appreciate the overall positive remarks by reviewer #3. Below we detail how the remaining concerns of this reviewer have been considered.

*1. The authors challenge *Nramp*^{+/+} CBA mice and look at acute responses, what is the rationale of looking at mast cells during this acute time point. Would the authors expect similar responses during a more chronic time point e.g. post 10 days of infection?*

Response: The use of CBA mice for the *in vivo* experiments is indeed because such *Nramp* proficient mice allow flexible analysis of *Salmonella* infections across time-frames and inoculum doses without the risk for lethality. In other parallel projects unrelated to this

manuscript, long-term chronic *Salmonella* infections are also under study in CBA mice by some of the authors (Sellin & colleagues, Fällman & colleagues; in progress).

This present study uses *Salmonella* Typhimurium (and the related *E. coli* and *Yersinia* strains) to probe how mast cells (MCs) respond to invasive versus extracellular enterobacteria. For *Salmonella* to be actively invasive, expression of type-three-secretion-system-1 (TTSS-1) and the cognate effectors is paramount. We know from prior work that expression of this machinery is turned on in the gut lumen during the acute phase of per-oral infection and sparks invasion of multiple cell types in the gut mucosa. This mucosal onslaught by TTSS-1 “ON” *Salmonella* continues for several days, although notably TTSS-1 “OFF” subpopulations also exist in this gut niche due to bistable TTSS-1 expression (¹). Importantly, as the infection progresses, *Salmonella* bacteria that translocate into deeper tissues turn off TTSS-1 expression and are therefore no longer actively invasive (e.g.²). There is also evidence that *Salmonella* TTSS-1 expression may over time be counter selected against in the lumen, leading to emergence of TTSS-1-inactivated mutants (¹). Finally, chronic *Salmonella* gut infections present with a more heterogenous mucosal tissue architecture, making microscopy-based analyses more challenging. For these reasons, we chose an early acute time point to explore MC coexistence with invasive *Salmonella in vivo* (Fig 1 and Fig 6).

Nevertheless, the reasoning above can be seen as theoretical. To substantiate this choice experimentally, we have now examined the mucosal tissue architecture, MC numbers and locations, and their relationship to *Salmonella* bacteria, also during later chronic infection in mice. The results reveal that i) MC numbers in the *Salmonella*-infected mucosa do not change dramatically between acute and chronic infection, that ii) *Salmonella* MC coexistence still is detectable also at later infection stages, but that iii) the infected tissue architecture, level of local *Salmonella* colonization, and consequently MC – *Salmonella* interactions appear more variable. In the revised manuscript, we have included example data from a stable chronic infection time-point (42 days post-infection) in Fig S1C-J. We have also introduced appropriate clarifications in the results text, comparing acute and chronic infection stages.

2. The authors discuss their findings in relation to previous findings saying that mast cells are found in the intestinal submucosa and mucosa prior to infection and that the population expands after infection. Except for expanding and being invaded, what do the authors propose that the infiltrating mast cells do in vivo?

In vitro the authors convincingly show that invading STm trigger the release of TNF. Based on previous studies of mast cells in the urinary tract, the Abraham lab have shown that mast cell derived TNF is essential for infiltrating neutrophil activation, do the authors propose a similar mechanism taking place in the cecum of STm infected mice? The same lab has also shown that mast cell derived IL-10 is essential for controlling bacterial burden, could this be a reason for the mast cells expansion in the mucosa in vivo?

Furthermore, this reviewer has concerns that the only read out of the mast cells in vivo is their number and their proximity to bacteria. What is the activation state of the mast cells? Do they have increased CD63 expression to signify degranulation? These questions could

easily be addressed by comparing neutrophil responses by flow cytometry or looking at epithelial integrity by FITC-dextran in WT vs mast cell deficient KitW-sh/W-sh mice during acute and more chronic infection stages. In addition, if the authors could show by flow cytometry that luminal and/or mucosal mast cells have STm inside would greatly support the *in vitro* findings of the paper and help distinguish whether these cells show signs of “slow” extracellular TLR4 activation or if this is due to intracellular activation.

Response: Thank you for these thought-provoking comments.

[2a response; on the subpoints regarding what the infected MCs do, and the roles of secreted TNF, IL-10 et.c. in relation to previous literature]: Our data from multiple MC models show that that the *Salmonella*-infected MC produce and secrete a wide range of inflammation-modulating cytokines (IL6, TNF, IL-10 et.c.) and chemokines (CCL2, CCL3 et.c.). Previous work has demonstrated that at least two major cellular responses, the intestinal epithelial inflammasome response (^{3,4}), and a macrophage TNF response (⁵) trigger the global initiation of acute mucosal tissue inflammation in the *Salmonella*-infected gut. From this, and the relatively modest numbers of MCs, it appears unlikely that the herein described MC response will be decisive for the global tissue inflammation state. Rather, we argue that *Salmonella*-stimulated MCs contribute to shaping the local cellular environment at mucosal infection sites. This conclusion gains concrete support from i) that we see dense aggregations of CD45+ and CD18+ cells around MCs in the infected mucosa in mice (Fig 6C), and ii) that supernatants from *Salmonella*-infected MCs indeed promote survival and differentiation of other immune cell subsets in our bone-marrow co-culture experiments (Fig 6D-F). This outcome would broadly be in line with the mentioned findings by the Abraham group under different infection conditions. We have elaborated on these conclusions in the discussion.

[2b response; regarding the MC activation state]: As suggested by the reviewer, we have now adapted a surface-CD63 staining protocol. This serves as a readout for activation/degranulation that complements the β -hexosaminidase assay. From the new results presented in Fig S2J-K, it is evident that treatment of murine MCs with the A23187 agonist (positive control) as anticipated generates a dramatic increase in CD63^{high} cells. By sharp contrast, no such effect was seen upon MC infection with either wild-type or $\Delta invG$ *Salmonella* for 1h, or for 24h (Fig S2J-K). Consequently, we also did not find CD63 staining of MCs in *Salmonella*-infected murine tissues informative. These observations, which are in full agreement with the β -hexosaminidase assay results (Fig S2F-I), demonstrate that *Salmonella* infection does not result in notable MC degranulation.

[2c response; on the possibility for further *in vivo* phenotyping and MC deletion]: We of course agree that such experimentation could in theory be exciting. However, as described in the “2a response” above, we already know that there are multiple cellular signaling pathways triggered in parallel during the acute *Salmonella* gut infection in mice. At least two of these, the epithelial cell NAIP/NLRC4 inflammasome response (^{3,4}) and a sentinel macrophage TNF response (⁵), fire promptly upon *Salmonella* invasion of the gut tissue and drive the first signs of global mucosal inflammation. This highly parallel wiring (which would require a multi-gene-knockout background mouse strain to isolate the MC contribution), and the fact that the MC specificity of KitW-sh/W-sh and similar MC ablation models has been called into question (⁶), prohibits meaningful further analysis in this direction. This is the reason why we have opted for the combination of infected gut tissue microscopy and a well-

controlled *ex vivo* co-culture assay (Fig 6C-F) to probe the functional consequence of *Salmonella*-induced MC secretions.

Finally, we would like to reiterate that the main point of our manuscript comprises the finding that MCs can combine effector-triggered immune signaling and classical PAMP sensing to differentiate their output between extracellular (weak cytokine output) and invasive (strong cytokine output) enterobacterial infection. We argue that this conclusion is both novel, well consolidated in the presented data, and with potentially broad-reaching ramifications for many different types of infections. Future studies across a range of pathogens and infection models will have to probe the specific physiological outcome(s) of this MC property in each particular case.

3. One of the strengths of the paper is that the authors translate findings from BMMCs and PCMCs to the human mast cell line and show some similar phenotypes between these systems. The paper would however improve further by validating some of the RNA-seq hits in the human in vitro system, and thereby showing that these findings are translatable between mouse and human cells.

Response: We agree that this is a relevant extension. Therefore, we have conducted additional transcript quantifications in the human MC model, guided by the results from the BMMC RNASeq analysis. Following challenge with invasive and non-invasive *Salmonella* strains, we find that the human MCs indeed upregulate cytokine/chemokine transcripts in a similar fashion as the murine BMMCs and PCMCs. Specifically, invasive *Salmonella* infection elicits vigorous production of *TNF*, *IL6*, *IL13*, *CCL3* and *CCL4* transcripts, whereas the non-invasive ($\Delta invG$) strain causes a much weaker transcript response. Further in line with the BMMCs and PCMCs results (see Fig S3), another set of transcripts that include *NLRP3* and *IL1 β* show pronounced upregulation in response to both strains. These supporting results are incorporated in Fig S7G of the revised manuscript and mentioned in the corresponding results section text. To simplify comparisons with the RNASeq results, we have presented these new qPCR data as normalized Z-scores in Fig S7G. The corresponding raw data can be found in the source data table.

4. The authors show that mast cells do not degranulate after 1h while they are transcriptionally active at this time point. Does this mean that all of the released cytokines synthesized de novo during the time of infection? The majority of the assays take place at 4 or 24 hours post infection, how do the β -hex levels compare at these time points?

Furthermore, is intracellular STm a necessity for degranulation? in that case investigating the kinetics of the uptake and correlate this to degranulation would support the authors point.

Response: To conclusively resolve this question, we have conducted several additional degranulation experiments, both at 4h and 24h post-infection. The results demonstrate that neither invasive, nor non-invasive *Salmonella* strains elicit any pronounced β -hexosaminidase release within 1-24h post-infection (new Fig S2G-H). Moreover, as advised by this reviewer (comment 3 above), we have also adopted a CD63 staining protocol as an alternative method to assess MC degranulation/activation. Consistent with the β -

hexosaminidase assay results, we again find that A23187 stimulation of BMMCs fuels the emergence of a CD63^{high} MC population, while *Salmonella* infection does not (data from 1 and 24h post-infection shown in the updated Fig S2J-K). By contrast, *Salmonella* infection leads to a near-instant induction of cytokine transcription in BMMCs, and the corresponding release of the formed cytokine proteins with a somewhat delayed kinetics (Fig S2E and Fig S2D). From this, we conclude that MC production of immunomodulatory cytokines and chemokines upon *Salmonella* exposure is a *de novo*-synthesis process, that does not depend on preformed mediator release through degranulation.

5. The authors convincingly show that STm invade mast cells, however, there is limited discussion to why STm would want to invade these cells. To this reviewer's knowledge, mast cells do not act as antigen presenting cells, nor migrate from inflamed tissue to peripheral sites, so how do Salmonella benefit from invading these cells? Investigating STm replication capability in mast cells using the gent-protection assay and plating CFUs or by microscopy/flow cytometry using the pFCcgi plasmid system would answer this question and greatly improve the knowledge of STm/mast cell interactions.

The authors further show that sopB activate AKT phosphorylation and discuss how this can affect downstream cytokine activation. In line with the previous comment, sopB has also been shown to phosphorylate AKT in B cells in a way to promote Salmonella survival and this should perhaps be discussed as a potential effect of the phenotype observed for sopB infected mast cells and not only that it may drive cytokine production.

Response: These are important points, and partially overlap with earlier thoughts by reviewer #2 (see response-to-reviewers letter for revision R1). We have indeed conducted such long-term infection experiments with an intracellular reporter *Salmonella* strain (*ssaG*-GFP), where Gentamicin is added to the medium and the infected MCs subsequently followed up to 72h post-infection. The results from these experiments can be found here below in the rebuttal letter (Fig R2). These data support that *Salmonella* can linger within MCs for several days (% *Salmonella* positive MCs appears unaltered over time; Fig R2A), but at the same time the expansion of the intracellular bacteria (judged by average GFP fluorescence per infected cell over time; Fig R2B) appears limited between 24-72h post-infection. In follow-up studies, we aim to investigate the intracellular *Salmonella* life style and replication potential within MCs at much greater detail, using a combination of high-resolution live- and fixed microscopy and bacterial genetics. We therefore want to avoid overly strong statements on this topic in the present manuscript. However, we have now further discussed that *Salmonella* can establish a stable intracellular population within MCs, and that this could potentially be a niche for long-term persistence. Moreover, we have commented further on the potential for SopB to increase the longevity of infected MCs, including citing of such prior observations in B cells.

Figure R2. A: MCs harboring vacuolar *S.Tm*/*psaG*-GFP+ persist for up to 72h across a range of MOIs. Quantification of BMMCs harboring vacuolar *S.Tm* (SL1344 background; *psaG*-GFP reporter). **B:** Median fluorescence intensities of *ssaG*-GFP+ MCs in A. The experiment was performed twice with a total of 4 replicates originating from distinct overnight cultures. Mean \pm SEM of pooled biological replicates are shown. Groups were statistically compared by two-way ANOVA and Dunnett's posthoc test, comparing 48 and 72h groups to 24h post-infection.

6. Since the authors eloquently show that there are impaired cytokine expression and release in MCs infected by the different mutants, do the authors believe that there would be a phenotype in mice as well? e.g. would there be less mucosal/luminal MCs than in SL1344WT infected mice?

Response: Our original observations from uninfected mice and *Salmonella*-infected mice at 48 hours post-infection (Fig 1), as well the new analyses during chronic infection stages (42 days post-infection; Fig S1C-J), show that MCs are present in the murine gut mucosa already before the infection, and increase approximately \sim 2-fold in infected tissue. Hence, the *Salmonella* infection appears to have only a modest impact on the total MC numbers. However, our data from transcript and protein quantification experiments in murine C57BL/6 BMMCs, C57BL/6J BMMCs, C57BL/6J *Tlr4*^{-/-} BMMCs, C57BL/6 PCMCs, and human LUVA cells (including also the new data in Fig S7G), all converge on the conclusion that effector-driven invasion fuels full-blown cytokine secretion from infected MCs. From this, it appears highly plausible that the local cytokine response from infected MCs *in vivo* will also be dictated by whether the engaging bacteria translocate TTSS effectors (two-step activation) or not (one-step activation). We have elaborated on this point in the discussion section.

Minor comments: The flow plots in Supplementary figure 5 are not properly compensated

Response: Well spotted. Due to the simple setup with only two fluorescence channels of modest overlap, we realize that insufficient attention had been placed on the compensation in this case. We have now conducted additional control experiments to optimize the compensation, which has led to an updated Fig S5. As evident from these flow panels, the optimized compensation only led to minor changes in the data point spread. We have also

reanalyzed the data in Fig 4B with these new settings. Finally, an additional infection and staining experiment was conducted and pooled in with the prior data for Fig 4B, to ensure reproducibility. This has led to subtle changes in the absolute numerical values of data presented in Fig 4B. However, the conclusions from these experiments remain identical.

We hope that the above described additions and adjustments satisfy the remaining concerns of reviewer #3. Additionally, we have again carefully pruned the manuscript for consistency and clarity, which has led to a few minor changes throughout the text. All revisions are visible in the marked-up version (yellow highlight) of the here revised manuscript. We thank all three reviewers again for their input and look much forward to your response.

With best regards,

The Authors

References to response letter

1. Diard, M. *et al.* Stabilization of cooperative virulence by the expression of an avirulent phenotype. *Nature* **494**, 353–356 (2013).
2. Hausmann, A. *et al.* Intestinal epithelial NAIP/NLRC4 restricts systemic dissemination of the adapted pathogen *Salmonella Typhimurium* due to site-specific bacterial PAMP expression. *Mucosal Immunol* **13**, 530–544 (2020).
3. Sellin, M. E. *et al.* Epithelium-Intrinsic NAIP/NLRC4 Inflammasome Drives Infected Enterocyte Expulsion to Restrict *Salmonella* Replication in the Intestinal Mucosa. *Cell Host & Microbe* **16**, 237–248 (2014).
4. Rauch, I. *et al.* NAIP-NLRC4 Inflammasomes Coordinate Intestinal Epithelial Cell Expulsion with Eicosanoid and IL-18 Release via Activation of Caspase-1 and -8. *Immunity* **46**, 649–659 (2017).
5. Hausmann, A. *et al.* Intercrypt sentinel macrophages tune antibacterial NF- κ B responses in gut epithelial cells via TNF. *Journal of Experimental Medicine* **218**, (2021).
6. Rodewald, H.-R. & Feyerabend, T. B. Widespread Immunological Functions of Mast Cells: Fact or Fiction? *Immunity* **37**, 13–24 (2012).

REVIEWERS' COMMENTS

Reviewer #3 (Remarks to the Author):

The authors have done a good job of addressing my concerns.

Uppsala
December 20th, 2023

Reviewer #3

Comment: *“The authors have done a good job of addressing my concerns.”*

Author response: We are happy to hear that reviewer #3 finds our revisions compelling.